# GRADIENT-BASED CONSTRAINED MULTI-OBJECTIVE OPTIMIZATION

## ABSTRACT

There is more and more attention on constrained multi-objective optimization (CMOO) problems, however, most of them are based on gradient-free methods. This paper proposes a constraint gradient-based algorithm for multi-objective optimization (MOO) problems based on multi-gradient descent algorithms. We first establish a framework for the CMOO problem. Then, we provide a Moreau envelope-based Lagrange Multiplier (MLM-CMOO) algorithm to solve the formulated CMOO problem, and the convergence analysis shows that the proposed algorithm convergence to Pareto stationary solutions with a rate of $\mathcal{O}(\frac{1}{\sqrt{T}})$. Finally, the MLM-CMOO algorithm is tested on several CMOO problems and has shown superior results compared to some chosen state-of-the-art designs.

## 1 INTRODUCTION

Multi-objective optimization (MOO) is widely used in many real-world application scenarios, such as, in online advertising, the models need to maximize both the Click-Through Rate and the Post-Click Conversion Rate. In MOO, one attempts to simultaneously optimize several, potentially conflicting functions. MOO has wide applications in all industry sectors where decision-making is involved due to the natural appearance of conflicting objectives or criteria. Applications span across applied engineering, operations management, finance, economics, and social sciences, agriculture, green logistics, and health systems. When the individual objectives are conflicting, no single solution exists that optimizes all of them simultaneously. In such cases, the goal of MOO is then to find Pareto optimal solutions (also known as efficient points), roughly speaking points for which no other combination of variables leads to a simultaneous improvement in all objectives. The determination of the set of Pareto optimal solutions helps decision makers to determine the best trade-offs among the several competing criteria.

MOO research can be divided into 2 categories, which are gradient-free and gradient-based methods. For the gradient-free method, people focus on evolutionary and Bayesian MOO algorithms, which are suitable for small-scale problems but less practical for high-dimensional MOO models and can not provide a convergence guarantee. On the contrary, the gradient-based method can provide a convergence guarantee in strongly convex, convex, and non-convex functions for MOO problems with different assumptions. The CMOO problem in the gradient-free method is well-developed. However, there is no gradient-based method for the CMOO problem.

Compared to conventional single-objective optimization, one key difference in MOO is the coupling and potential conflicts between different objective functions. As a result, there may not exist a common solution that minimizes all objective functions. Rather, the goal in MOO is to find a *Pareto stationary solution* that is not improvable for all objectives without sacrificing some objectives. The gradient-based method MOO has 2 lines, single-objective transformation, and the conflict gradients alleviating method, where the latter has garnered more attention in recent years due to their better performances. The single-objective transformation is the first step for the gradient-based method MOO method. It first transfers a MOO problem into a single-objective optimization (SOO) problem with a given fixed coefficient. With the sufficient algorithm in SOO, it is easy to solve. However, this transformation can not give a stable guarantee for the convergence rate as it may give the farthest Pareto front for the given coefficient. Then, the conflict gradients alleviating method is proposed to resolve the conflicting gradients in MOO. However, none of them pay attention to the gradient-based CMOO problem.

Although many gradient-free methods can provide solutions to CMOO problems, very few of them can provide the convergence guarantee. In addition, gradient-free methods are more suitable for small-scale and low-dimension MOO problems, which limits the application of gradient-free algorithms. Thus, we provide a Moreau envelope-based Lagrange Multiplier (MLM-CMOO) algorithm for the CMOO problem via the gradient method. Our contributions are summarized as follows.

- We propose MLM-CMOO, which solves the CMOO problem using the gradient-based method. MLM-CMOO first divides the CMOO problem into 2 parts, maximizes the minimum decrease across the losses, and makes the decrease obey the constraints. To maximize the minimum decrease across the losses, we use a similar method in MGDA (Sener & Koltun, 2018). To limit the decrease, we the Moreau envelope-based proximal gradient method.

- We provide convergence analyses for MLM-CMOO with convex multi-objectives and convex multi-constraints. The convergence rate of MLM-CMOO is $\mathcal{O}(\frac{1}{\sqrt{T}})$

- We conduct numerical experiments to verify the effectiveness of MLM-CMOO. The experimental results demonstrate the efficiency of the MLM-CMOO.

The remainder of this paper is organized as follows. Section 2 reviews related work. In Section 3, we present the system model and algorithm design of MLM-CMOO. In Section 4, we provide the convergence analysis of the MLM-CMOO algorithm. Numerical results and conclusions are provided in Section 5 and Section 6, respectively.

## 2 RELATED WORK

**MOO.** MOO algorithms can be grouped into two main categories. The first line of work is gradient-free methods (e.g., evolutionary MOO algorithms and Bayesian MOO algorithms (Lin et al., 2022; Zhang & Li, 2007; Laumanns & Ocenasek, 2002; Deb et al., 2002; Belakaria et al., 2020)). These methods are more suitable for small-scale problems but less practical for high-dimensional MOO models (e.g., deep neural networks). (Do et al., 2023; Zheng et al., 2022) provided the convergence analysis for gradient-free methods to solve MOO problems. The second line of work focuses on gradient-based approaches (Liu & Vicente, 2024; J. Fliege & Vicente, 2019; Momma et al., 2022; Peitz & Dellnitz, 2018; Désidéri, 2012), which are more practical for high-dimensional MOO problems. However, while having received increasing attention from the community in recent years, the Pareto-stationary convergence analysis of these gradient-based MOO methods attracts much more attention.

Various works explored the convergence rates under different assumptions in strongly convex, convex, and non-convex functions for MOO problems. Using full gradient, MGD (J. Fliege & Vicente, 2019) could achieve tight convergence rates in strongly-convex and non-convex cases, i.e., linear rate $\mathcal{O}(r^T)$, $r \in (0, 1)$ and sub-linear rate $\mathcal{O}(1/T)$. However, it requires a linear search of the learning rate in the algorithm and sequence convergence ($\{x_t\}$ converges to $x^*$). The linear search of learning rate is a classic technique but does not fit in gradient-based algorithms in deep learning. Moreover, the sequence convergence assumption is too strong. If using a stochastic gradient, SMGD (Liu & Vicente, 2024) methods make a further complicated case. The stochastic gradient noise would complicate the analysis. an $\mathcal{O}(1/T)$ rate analysis for SMGD was provided in (Liu & Vicente, 2024) based on rather strong assumptions on a first-moment bound and Lipschtiz continuity of common descent direction. On the other hand, (Liu & Vicente, 2024) and (Zhou et al., 2022) showed that the common descent direction provided by the SMGD method is likely to be a biased estimation, which may cause divergence issues. Recently, by utilizing momentum, oCo (Fernando et al., 2024) and CR-MOGM (Zhou et al., 2022) were proposed with corresponding convergence guarantees. (Xiao et al., 2023) utilized a direction-oriented approach by a preference direction. (Yang et al., 2023) proposed a federated MOO algorithm with GD and SGD matching previous centralized MOO algorithms.

**CMOO.** Most existing CMOO research focuses on gradient-free methods. SaE-CMO (Song et al., 2024) proposed a cooperative evolutionary algorithm with a dual-population approach to enhance search progress. PAC-MOO was proposed in (Ahmadianshalchi et al., 2024) based on Bayesian optimization. (Zhang et al., 2024) introduced a dynamic assistant population to search direction for CMOO. (Yang et al., 2024) proposed a feasibility tracking strategy to explore all feasible regions

for CMOO. (Belaiche et al., 2023) proposed PCMOEA/D-DMA based on a multi-population mechanism and implemented under a synchronous master-slave parallel model to select the best Pareto front based on an elitism mechanism. (Li et al., 2023) proposed a surrogate-ensemble-assisted co-evolutionary algorithm to improve the search efficiency.

**Constraint Handling Techniques.** Penalty function methods, decoders, special operators, and separation techniques are a simple taxonomy of the constraint handling methods in nature-inspired optimization algorithms. There are several types of penalty functions used with evolutionary algorithms (EAs), the most important ones include (Kramer, 2010) Death penalty, Dynamic penalty, Static penalty, Adaptive penalty, and Stochastic ranking. As an example of decoders, (Koziel & Michalewicz, 1998) proposed a homomorphous mapping (HM) method between an n-dimensional cube and feasible space. The feasible region can be mapped onto a sample space where a population-based algorithm could run a comparative performance (Koziel & Michalewicz, 1998; Kim & Husbands, 1998a;b; Koziel & Michalewicz, 1999). However, this method requires high computational costs. A special operator is used to preserve the feasibility of a solution or move within a special region (Michalewicz, 1996; Schoenauer & Michalewicz, 1996; 1997). Nevertheless, this method is hindered by the initialization of feasible solutions in the initial population, which is challenging with highly constrained optimization problems. Unlike the penalty function technique, another approach separates the values of objective functions and constraints in the nature-inspired algorithms (NIAs) (Powell & Skolnick, 1993), which is known as the separation of objective function and constraints. The authors of (Hinterding & Michalewicz, 1998) initially proposed dividing the search space into two phases. In the first phase, feasible solutions are found, and optimizing the objective function is considered in the second phase. Representative methods of this type of CHT are the Constraint dominance principle (CDP), Epsilon CHT, and Feasibility rules.

## 3 SYSTEM AND PROBLEM FORMULATION

This section introduces the basic background knowledge of MOO, typical algorithms, and their convergence analysis. Then, we proposed the objective of CMOO and the algorithm to solve the formulated problem.

### 3.1 MULTI-OBJECTIVE OPTIMIZATION

Multi-objective optimization (MOO) is concerned with solving the problems of optimizing multi-objective functions simultaneously, which can be formulated as

$$\min_{x} F(x) = (f_1(x), f_2(x), ..., f_n(x))^\top ,\tag{1}$$

where $f_i$ are real-valued functions, and $\mathcal{N}$ represents the set of the total $n$ objectives ($n > 2$). The MOO problem is smooth if all objective functions $f_i$ are continuously differentiable. Different from single objective optimization where 2 solutions $x$, $y$ can be ordered by $f(x) < f(y)$ or $f(x) \geq f(y)$. MOO could have two parameter vectors where one performs better for task $i$ and the other performs better for task $j$, where $i \neq j$. Therefore, Pareto optimality is defined to deal with such an incomparable case.

**Definition 3.1** *(Pareto optimality). For any two solutions $x_1$, $x_2 \in \mathcal{X}$, we say that $x_1$ dominates $x_2$, denoted as $x_1 \prec x_2$, if $f_i(x_1) \leq f_i(x_2)$ for all $i$, and there exists one $i$ such that $f_i(x_1) < f_i(x_2)$; otherwise, we say that $x_1$ does not dominate $x_2$, denoted as $x_1 \nprec x_2$. A solution $x^* \in \mathcal{X}$ is called Pareto optimal if it is not dominated by any other solution in $\mathcal{X}$.*

Note that a set of Pareto optimal solutions is called a Pareto set. The goal of MOO is to find a Pareto optimal solution, which must be Pareto critical (Custódio et al., 2011).

**Definition 3.2** *(Pareto criticality). A solution $x^* \in \mathcal{X}$ is called Pareto critical if there is no common descent direction $\boldsymbol{d}$ such that $\nabla f_i(x^*)^\top \boldsymbol{d} < 0$, $i \in \mathcal{M}$ for all objectives.*

This definition indicates that if $x$ is not Pareto critical, such direction $\boldsymbol{d}$ will be a local descent direction for $\boldsymbol{F}$ at point $x$. Optimizing through $\boldsymbol{d}$ in the local neighborhood of $x$ can get a better solution that dominates $x$ (J. Fliege & Vicente, 2019). Since Pareto criticality reflects the local

property compared with Pareto optimality, it is often used as the local minimal condition for MOO with non-convex objectives (Fliege & Svaiter, 2000). We then present sufficient conditions for determining Pareto criticality/optimality, which appear as metrics to study the convergence for the MOO algorithm (Fliege & Svaiter, 2000; H. Tanabe & Yamashita, 2023).

Similar to single-objective optimization, MOO can be solved by running iteratively with gradient-based algorithms. For example, in MGDA (Sener & Koltun, 2018), it directly optimizes towards the Pareto criticality in Definition 3.2. Specifically, in each iteration, MGDA aims to find a direction $\boldsymbol{d}$ to maximize the minimum decrease across the losses by solving the following subproblem,

$$\max_{\boldsymbol{d}} \min_i (f_i(x) - f_i(x + \eta\boldsymbol{d})) \approx \eta \max_{\boldsymbol{d}} \min_i \nabla f_i(x)^\top \boldsymbol{d}.$$

By regularizing the norm of $\boldsymbol{d}$ on the right side, it computes the direction by

$$\boldsymbol{d} = \arg\min_{\boldsymbol{d}} \{ \max_i \lambda_i \nabla f_i(x) + \frac{1}{2} \|\boldsymbol{d}\|^2 \}.$$

This sub-problem can be rewritten equivalently as the following differentiable quadratic optimization

$$\boldsymbol{d}, \mu = \arg\min_{\boldsymbol{d}, \mu} (\frac{1}{2} \|\boldsymbol{d}\|^2 + \mu), \ s.t. \ \lambda_i \nabla f_i(x) \leq \mu.$$

If $\mu < 0$, then $\nabla f_i(x)^\top \boldsymbol{d} < 0$, which means $x$ is not Pareto critical from Definition 3.2, and $\boldsymbol{d}$ is the direction to descent all the objectives simultaneously (Fliege & Svaiter, 2000; J. Fliege & Vicente, 2019). To simplify the optimization, such a primal problem has a dual objective as a min-norm oracle

$$\boldsymbol{\lambda} = \arg\min_{\boldsymbol{\lambda}} \left\| \sum_{i=1}^m \lambda_i \nabla f_i(x) \right\|.$$

The direction is then calculated by $\boldsymbol{d} = -\sum_{i=1}^m \lambda_i \nabla f_i(x)$.

**Convergence analysis**. MGDA has been shown to converge to an arbitrary Pareto critical/optimal point with the same rate as single-objective optimization (J. Fliege & Vicente, 2019). A similar result has been proved with PCGrad (Yu et al., 2020). CAGrad has been shown to converge to the minimizer or stationary point of the averaging loss $\frac{1}{n}\sum_{i=1}^n f_i(x)$ when $c \in [0, 1)$, or an arbitrary Pareto critical/optimal point when $c \geq 1$ (Liu et al., 2021).

### 3.2 CONSTRAINED MULTI-OBJECTIVE OPTIMIZATION

Followed by previous research, this paper considers a MOO problem, where each objective has its constraints. It is formulated as

$$\min_x F(x) = (f_1(x), f_2(x), ..., f_n(x))^\top,$$
$$s.t. \ g_i(x) \leq 0, \ \forall i \in \mathcal{N}.$$

Similar to MGDA, we aim to find a direction $\boldsymbol{d}$ to maximize the minimum decrease across the losses. Thus, our problem can be reformulated as

$$\min_x F(x, \boldsymbol{\lambda}) := \sum_{i=1}^m \lambda_i f_i(x), \tag{2}$$

$$s.t. \ \boldsymbol{\lambda} = \arg\min_{\boldsymbol{\lambda}'} \left\| \sum_{i=1}^m \lambda_i' \nabla f_i(x) \right\|,$$

$$g_i(x) \leq 0, \ \forall i \in \mathcal{N}.$$

To simiplify the expression, we note $H(x, \boldsymbol{\lambda}) := \|\sum_{i=1}^m \lambda_i \nabla f_i(x)\|$, due to the non-smooth of $H(x, \boldsymbol{\lambda})$, we use $\nabla H(x, \boldsymbol{\lambda})$ to express the proxmial gradient of $H(x, \boldsymbol{\lambda})$, where $\nabla_x H(x, \boldsymbol{\lambda}) := \arg\min_u \{ H(u, \boldsymbol{\lambda}) + \frac{1}{2} \|u - x\|^2 \}$ and $\nabla_{\boldsymbol{\lambda}} H(x, \boldsymbol{\lambda}) := \arg\min_v \{ H(x, \boldsymbol{v}) + \frac{1}{2} \|\boldsymbol{v} - \boldsymbol{\lambda}\|^2 \}$.

To solve the above problem 2, we first find the suitable coefficient ($\boldsymbol{\lambda}$) of the problem 2, where we get a subproblem.

$$\min_{\boldsymbol{\lambda}} H(x, \boldsymbol{\lambda}), \tag{3}$$
$$s.t. \ g_i(x) \leq 0, \ \forall i \in \mathcal{N}.$$

Then, we transfer problem 3 into an unconstrained optimization problem via the Lagrange Multiplier method, which is expressed as

$$L = H(x, \boldsymbol{\lambda}) + \sum_{i=1}^{m} \mu_i g_i(x). \tag{4}$$

Due to the absolute value of $\sum_{i=1}^{m} \lambda_i \nabla f_i(x)$, the Lagrange function ($L$) may not be smooth, thus we use a Moreau envelope-based Lagrange Multiplier function to solve above problems, which can be expressed as

$$L_s(x, \boldsymbol{\lambda}, \boldsymbol{z}) := \min_{\theta} \max_{\boldsymbol{\mu}} \left\{ H(x, \boldsymbol{\theta}) + \sum_{i=1}^{N} \mu_i g_i(x) + \frac{1}{2\gamma_1} \sum_{i=1}^{N} \|\theta_i - \lambda_i\|^2 - \frac{1}{2\gamma_2} \sum_{i=1}^{N} \|z_i - \mu_i\|^2 \right\},$$

where $\gamma_1, \ \gamma_2$ are the proximal parameter and $\gamma_1 \geq 0, \ \gamma_2 \geq 0$.

Employing the function of $L_s$, we reformulated the problem 2 as

$$\min_{x} F(x, \boldsymbol{\lambda}), \tag{5}$$
$$s.t. \ H(x, \boldsymbol{\lambda}) - L_s \leq 0.$$

To guarantee the theoretical convergence of the proposed method, instead of directly solving reformulation 5, we consider its variant using a truncated Lagrangian function,

$$L_{s,r}(x, \boldsymbol{\lambda}, \boldsymbol{z}) := \min_{\theta} \max_{\boldsymbol{\mu} \in Z} \left\{ H(x, \boldsymbol{\theta}) + \sum_{i=1}^{N} \mu_i g_i(x) + \frac{1}{2\gamma_1} \sum_{i=1}^{N} \|\theta_i - \lambda_i\|^2 - \frac{1}{2\gamma_2} \sum_{i=1}^{N} \|z_i - \mu_i\|^2 \right\}.$$

where $Z := [0, r]^p$ and $r > 0$. We define $\boldsymbol{\theta}^* := \boldsymbol{\theta}^*(x, \boldsymbol{\lambda}, \boldsymbol{z})$ and $\boldsymbol{\mu}^* := \boldsymbol{\mu}^*(x, \boldsymbol{\lambda}, \boldsymbol{z})$ is the unique saddle point of the above min-max problem. Compared with $L_s(x, \boldsymbol{\mu})$, the truncated version $L_{s,r}(x, \boldsymbol{\mu})$ is defined by maximizing $z$ over a bounded set $Z$. The truncated Lagrangian value function gives us the following variant to a reformulation of problem 5

$$\min_{x} F(x, \boldsymbol{\lambda}), \tag{6}$$
$$s.t. \ H(x, \boldsymbol{\lambda}) - L_{s,r} \leq 0.$$

Note that $\|\sum_{i=1}^{m} \lambda_i \nabla f_i(x)\| - L_s \leq 0$ for any $x, \ \boldsymbol{\lambda}$ in their domain. If $r$ is sufficiently large, the solution of reformulation of problem 5 can be obtained by solving variant problem 6. A comprehensive proof is presented in Theorem A.2 within Appendix A.3.

Then, problem 6 is solved by introducing a penalty parameter $\{c^{(t)}\}_{t=0}^{T-1}$, where $t$ is the round index,

$$\min_{x, \boldsymbol{\lambda}} \frac{1}{c^{(t)}} F(x, \boldsymbol{\lambda}) + H(x, \boldsymbol{\lambda}) - L_{s,r}.$$

The detailed steps are provided in Algorithm 1.

## 4 CONVERGENCE ANALYSIS

### 4.1 CONVERGENCE RESULTS OF MLM-CMOO

**Assumption 4.1** *In the CMOO, suppose the objectives $f_i$ and the constraints $g_i$ are convex for any $i \in \mathcal{N}$.*

---

**Algorithm 1** Moreau envelope-based Lagrange Multiplier Constrained MOO (MLM-CMOO)

---

1: **Input:** Initial point $x^{(0)}. \boldsymbol{\lambda}^{(0)}, \theta^{(0)}. \boldsymbol{\mu}^{(0)}$, penalty parameter $\{c^{(t)}\}_{t=0}^{T-1}$;
2: **for** $t = 0$ to $T - 1$ **do**
3:    $\boldsymbol{\theta}^{(t+1)} = \boldsymbol{\theta}^{(t)} - \eta \nabla_\theta L_{s,r}(x^{(t)} \boldsymbol{\lambda}^{(t)}, \boldsymbol{z}^{(t)})$;
4:    $\boldsymbol{\mu}^{(t+1)} = \boldsymbol{\mu}^{(t)} - \eta \nabla_\mu L_{s,r}(x^{(t)} \boldsymbol{\lambda}^{(t)}, \boldsymbol{z}^{(t)})$;
5:    $U = \arg\min_u \{H(u, \boldsymbol{\lambda}^{(t)}) + \frac{1}{2} \|u - x^{(t)}\|^2\} - \arg\min_u \{H(u, \boldsymbol{\theta}^{(t+1)}) + \frac{1}{2} \|u - x^{(t)}\|^2\}$;
6:    $x^{(t+1)} = x^{(t)} - \alpha \left( \frac{1}{c^{(t)}} \nabla_x F(x^{(t)}, \boldsymbol{\lambda}^{(t)}) + U + \sum_{i=1}^n \mu_i^{(t+1)} \nabla_x g_i(x^{(t)}) \right)$;
7:    $\boldsymbol{\lambda}' = \arg\min_{\boldsymbol{\lambda}} \|\sum_{i=1}^m \lambda_i \nabla f_i(x^{(t)})\|$;
8:    $\boldsymbol{V} = \arg\min_v \{H(x^{(t)}, \boldsymbol{v}) + \frac{1}{2} \|\boldsymbol{v} - \boldsymbol{\lambda}^{(t)}\|^2\} - \arg\min_v \{H(u, \boldsymbol{\theta}^{(t+1)}) + \frac{1}{2} \|u - x^{(t)}\|^2\}$;
9:    $\boldsymbol{\lambda}^{(t+1)} = \boldsymbol{\lambda}^{(t)} - \alpha \left( \frac{1}{c^{(t)}} (\boldsymbol{\lambda}^{(t)} - \boldsymbol{\lambda}') + \boldsymbol{V} - \frac{1}{\gamma_1} (\boldsymbol{\lambda})^{(t)} - \theta)^{(t+1)} \right)$;
10:   $\boldsymbol{z}^{(t+1)} = \boldsymbol{z}^{(t)} + \frac{\beta}{\gamma_2} (\boldsymbol{\mu}^{(t+1)} - \boldsymbol{z}^{(t)})$
11: **end for**

---

**Assumption 4.2** *For the general MOO, there exists a finite constant $B \in \mathbb{R}$, such that $0 \le \lambda_i^{(t)} \le B, \sum_{i=1}^N \lambda_i^{(t)} = 1$, for all $t = 0, ..., T - 1$.*

**Assumption 4.3** *$f_1(x), ..., f_N(x)$ are all differentialable, $S_f$-Lipschitz and $L_f$-smoothness, suggesting that for all $x$, $y$ and $i \in \mathcal{N}$, it holds $\|\nabla f_i(x)\| \le S_f$ and $\|\nabla f_i(x) - \nabla f_i(y)\| \le L_f \|x - y\| \le 2S_f$.*

**Assumption 4.4** *$g_1(x), ..., g_N(x)$ are all differentialable, $S_g$-Lipschitz and $L_g$-smoothness, suggesting that for all $x$, $y$ and $i \in \mathcal{N}$, it holds $\|\nabla f_i(x)\| \le S_g$ and $\|\nabla f_i(x) - \nabla f_i(y)\| \le L_g \|x - y\| \le 2S_g$.*

We first define a new function and then demonstrate the decreasing properties of this new function to show the convergence rate of the proposed algorithm.

$$V_t := \phi_{c_t}(x^{(t)}, \boldsymbol{\lambda}^{(t)}, \boldsymbol{z}^{(t)}) + C_{\theta,\mu} \left\| (\boldsymbol{\theta}^{(t)}, \boldsymbol{\mu}^{(t)}) - (\boldsymbol{\theta}^*(x^{(t)}, \boldsymbol{\lambda}^{(t)}, \boldsymbol{z}^{(t)}), \boldsymbol{\mu}^*(x^{(t)}, \boldsymbol{\lambda}^{(t)}, \boldsymbol{z}^{(t)})) \right\|^2,$$

where $C_{\theta,\mu} := \max\{(L_g + C_Z L_g)^2 + 1/(2\gamma_1^2) + L_g^2, 1/\gamma_2^2\}$, and $\phi_{c_t}(x^{(t)} \boldsymbol{\lambda}^{(t)}, \boldsymbol{z}^{(t)}) := \frac{1}{c^{(t)}} (F(x, \boldsymbol{\lambda}) - F^*) + H(x, \boldsymbol{\lambda}) - L_{s,r}$, where $F^*$ is the optimal value of function $F(x, \boldsymbol{\lambda})$.

**Lemma 4.5** *Under Assumptions 4.2, 4.3 and 4.4 hold, let $\gamma_1 \in (0, 1/\rho_T)$, $\gamma_2 > 0$, $c_t \le c_{t+1}$ and $\eta_t \in (\eta, \rho_\gamma/L_B^2)$ with $\eta > 0$, then there exist constants $c_\alpha, c_\beta > 0$ such that when $0 < \alpha \le c_\alpha$ and $0 < \beta \le c_\beta$, the sequence of $(x^{(t)}, \boldsymbol{\lambda}^{(t)}, \boldsymbol{z}^{(t)})$ generated by Algorithm 1: MLM-CMOO satisfies*

$$V_{t+1} - V_t \le -\frac{1}{4\alpha} \left\| x^{(t+1)} - x^{(t)} \right\|^2 - \frac{1}{4\alpha} \left\| \boldsymbol{\lambda}^{(t+1)} - \boldsymbol{\lambda}^{(t)} \right\|^2 - \frac{1}{4\beta} \left\| \boldsymbol{z}^{(t+1)} - \boldsymbol{z}^{(t)} \right\|^2$$

$$- \eta \rho_T C_{\theta,\mu} \left\| (\boldsymbol{\theta}^{(t)}, \boldsymbol{\mu}^{(t)}) - (\boldsymbol{\theta}^*(x^{(t)}, \boldsymbol{\lambda}^{(t)}, \boldsymbol{z}^{(t)}), \boldsymbol{\mu}^*(x^{(t)}, \boldsymbol{\lambda}^{(t)}, \boldsymbol{z}^{(t)})) \right\|^2.$$

The step sizes are carefully chosen to guarantee the sufficient descent property of $V_t$. This is essential for the non-asymptotic convergence analysis.

Given the decreasing property of $V_t$, we establish the non-asymptotic convergence analysis. The standard KKT conditions are inappropriate as necessary optimality conditions for problem equation 6. Motivated by the approximate KKT condition presented by (Andreani et al., 2010), which is characterized as an optimality condition for nonlinear program, regardless of constraint qualifications' fulfillment, we consider the following residual function $R_t := R_t(x^{(t)}, \boldsymbol{\lambda}^{(t)}, \boldsymbol{z}^{(t)})$ as a stationarity measure, we define the residual function as $R_t := dist(0, (\nabla F(x^{(t)}, \boldsymbol{\lambda}^{(t)}), 0)) + c_t((\nabla H(x^{(t)}, \boldsymbol{\lambda}^{(t)}), 0) - \nabla L_{s,r}(x^{(t)}, \boldsymbol{\lambda}^{(t)}, \boldsymbol{z}^{(t)})) + \mathcal{M}_{C \times Z}(x^{(t)}, \boldsymbol{\lambda}^{(t)}, \boldsymbol{z}^{(t)}))$, where $\mathcal{M}_\Omega(s)$ denotes the normal cone to $\Omega$ at $s$. This residual function $R_t$ also serves as a stationarity measure for the penalized problem of equation 6, with $c_t$ serving as the penalty parameter,

$$\min_{x, \boldsymbol{\lambda}, \boldsymbol{z} \in Z} \phi_{c_t}(x^{(t)}, \boldsymbol{\lambda}^{(t)}, \boldsymbol{z}^{(t)}) := F(x^{(t)}, \boldsymbol{\lambda}^{(t)}) + c_t \left( H(x^{(t)}, \boldsymbol{\lambda}^{(t)}) - L_{s,r}(x^{(t)}, \boldsymbol{\lambda}^{(t)}, \boldsymbol{z}^{(t)}) \right). \quad (7)$$

Evidently, $R_t = 0$ if and only if $(x^{(t)}, \boldsymbol{\lambda}^{(t)}, \boldsymbol{z}^{(t)})$ is a stationary point for the problem equation 7, meaning $0 \in \nabla \phi_{c_t}(x^{(t)}, \boldsymbol{\lambda}^{(t)}, \boldsymbol{z}^{(t)}) + \mathcal{M}_{C \times Z}(x^{(t)}, \boldsymbol{\lambda}^{(t)}, \boldsymbol{z}^{(t)})$.

**Theorem 4.6** *If Assumptions of Assumptions 4.2, 4.3 and 4.4 hold, let $\gamma_1 \in (0, 1/\rho_\gamma)$, $\gamma_2 > 0$, $c_t = \underline{c}(t+1)^p$ with $p \in (0, 1/2)$ and $\underline{c} > 0$. Pick $\eta_t \in (0, \rho_\gamma / L_B^2)$, then there exists $c_\alpha, c_\beta > 0$ such that when $\alpha \in (\underline{\alpha}, c_\alpha)$ and $\beta \in (\underline{\beta}, c_\beta)$, with $\underline{\alpha}, \underline{\beta} > 0$, the sequence of $(x^{(t)}, \boldsymbol{\lambda}^{(t)}, \boldsymbol{z}^{(t)}, \boldsymbol{\theta}^{(t)}, \boldsymbol{\mu}^{(t)})$ generated by Algorithm 1: MLM-CMOO satisfies*

$$\min_t \left\| (\boldsymbol{\theta}^{(t)}, \boldsymbol{\mu}^{(t)}) - (\boldsymbol{\theta}_r^*(x^{(t)}, \boldsymbol{\lambda}^{(t)}, \boldsymbol{z}^{(t)}), \boldsymbol{\mu}_r^*(x^{(t)}, \boldsymbol{\lambda}^{(t)}, \boldsymbol{z}^{(t)})) \right\| = \mathcal{O}(\frac{1}{\sqrt{T}}),$$

*and*

$$\min_t R_t(x^{(t)}, \boldsymbol{\lambda}^{(t)}, \boldsymbol{z}^{(t)}) = \mathcal{O}(\frac{1}{\sqrt{T^{1-2p}}}).$$

**Remark 1:** Theorem 4.6 first shows that the reformulated Lagrange Multiplier function $L_{s,r}(x, \boldsymbol{\lambda}, \boldsymbol{z})$ reaches its KKT stationary point with a convergence rate of $\mathcal{O}(\frac{1}{\sqrt{T}})$, which matches general constrained optimization method. In addition, MLM-CMOO can converge to the stationary point of the problem equation 7, where problem equation 7 is the penalized form of the original problem equation 2. Furthermore, the last statement shows that the convergence rate of each client's truncated proximal Lagrangian value function is related to the selection of the penalized parameter. The maximum convergence rate is $\mathcal{O}(\frac{1}{\sqrt{T}})$.

## 4.2 PROOF SKETCH

**Lemma 4.7** *Suppose the assumption of 4.3 and 4.4 hold, and let $\gamma_1 \in (0, 1/\rho_g)$, $\gamma_2 > 0$. Pick $\eta_t \in (0, \rho_T / L_B^2)$ with $L_B := \max\{(2 + C_z)L_g + 1/\gamma_1, L_g + 1/\gamma_2\}$ then the sequence of $(x^{(t)}, \boldsymbol{\lambda}^{(t)}, \boldsymbol{z}^{(t)}, \boldsymbol{\theta}^{(t)}, \boldsymbol{\mu}^{(t)})$ generated by Algorithm 1: MLM-CMOO satisfies*

$$\left\| (\boldsymbol{\theta}^{(t+1)}, \boldsymbol{\mu}^{(t+1)}) - \left( \boldsymbol{\theta}^*(x^{(t)}, \boldsymbol{\lambda}^{(t)}, \boldsymbol{z}^{(t)}), \boldsymbol{\mu}^*(x^{(t)}, \boldsymbol{\lambda}^{(t)}, \boldsymbol{z}^{(t)}) \right) \right\|$$

$$\leq (1 - \eta \rho_T) \left\| (\boldsymbol{\theta}^{(t)}, \boldsymbol{\mu}^{(t)}) - \left( \boldsymbol{\theta}^*(x^{(t)}, \boldsymbol{\lambda}^{(t)}, \boldsymbol{z}^{(t)}), \boldsymbol{\mu}^*(x^{(t)}, \boldsymbol{\lambda}^{(t)}, \boldsymbol{z}^{(t)}) \right) \right\|$$

Lemma 4.7 shows that the decay rate of the reformulated Lagrange Multiplier function $L_{s,r}(x, \boldsymbol{\lambda}, \boldsymbol{z})$ is related to the step size and problem parameters $((1 - \eta_t \rho_T))$. If we select the proper step size (i.e., $\eta_t \in (0, \rho_\gamma / L_B^2)$), the reformulated Lagrange Multiplier function converges to the KKT stationary point.

**Lemma 4.8** *Suppose the assumption of 4.2, 4.3 and 4.4 hold, and let $\gamma_1 \in (0, 1/\rho_g)$, $\gamma_2 > 0$. Pick $\eta \in (0, \rho_\gamma / L_B^2)$ with $L_B := \max\{2L_g + C_z L_g + 1/\gamma_1, L_g + 1/\gamma_2\}$ then the sequence of $(x^{(t)}, \boldsymbol{\lambda}^{(t)}, \boldsymbol{z}^{(t)})$ generated by Algorithm 1: MLM-CMOO satisfies*

$$\phi_{c_t}(x^{(t+1)}, \boldsymbol{\lambda}^{(t+1)}, \boldsymbol{z}^{(t+1)}) \leq \phi_{c_t}(x^{(t)}, \boldsymbol{\lambda}^{(t)}, \boldsymbol{z}^{(t)}) - (\frac{1}{2\beta} - \frac{L_{v_z}}{2}) \left\| \boldsymbol{z}^{(t+1)} - \boldsymbol{z}^{(t)} \right\|^2$$

$$- \left( \frac{1}{2\alpha} - \frac{L_{\phi_k}}{2} - \frac{\beta L_{\theta,\mu}^2}{\gamma_2^2} \right) \left( \left\| x^{(t+1)} - x^{(t)} \right\|^2 + \left\| \boldsymbol{\lambda}^{(t+1)} - \boldsymbol{\lambda}^{(t)} \right\|^2 \right)$$

$$+ \frac{\alpha}{2} \left( 2(L_g + C_z L_g)^2 + \frac{1}{\gamma_1^2} \right) \left\| \boldsymbol{\theta}^{(t+1)} - \boldsymbol{\theta}^*(x^{(t)}, \boldsymbol{\lambda}^{(t)}, \boldsymbol{z}^{(t)}) \right\|^2$$

$$+ (\alpha L_g^2 + \frac{\beta}{\gamma_2^2}) \left\| \boldsymbol{\mu}^{(t+1)} - \boldsymbol{\mu}^*(x^{(t)}, \boldsymbol{\lambda}^{(t)}, \boldsymbol{z}^{(t)}) \right\|^2.$$

*where $L_{\phi_t} := L_f / c_t + L_g + \rho_v$.*

Lemma 4.8 shows that a client's Lagrangian reformulation of objects decreases with the distance between its current parameters $(x^{(t)}, \boldsymbol{\lambda}^{(t)}, \boldsymbol{z}^{(t)})$. In addition, the bias of the reformulated Lagrange Multiplier function affects the convergence rate. The more accurate the reformulated Lagrange

Multiplier function is, the faster the reformulated Lagrangian reformulation of CMOO objectives decreases. Then, the proof of Lemma 4.8 lies in 2 parts: the descent of the MOO objectives ($F(x, \boldsymbol{\lambda})$) and the descent of the reformulated Lagrange Multiplier function $L_{s,r}(x, \boldsymbol{\lambda}, \boldsymbol{z})$. The challenge of the proof lies in 2 aspects: the decreasing of the MOO objectives based on dynamic coefficient ($\boldsymbol{\lambda}$) and binding the reformulated Lagrange Multiplier function $L_{s,r}(x, \boldsymbol{\lambda}, \boldsymbol{z})$.

## 5 EXPERIMENTS

This section introduces the experimental setups, including the datasets and models, multi-objective optimization algorithm setup, and experimental settings. We use multi-task learning experiments to verify the effectiveness of the proposed method. A typical MTL system is given a collection of input points and sets of targets for various tasks per point. A common way to set up the inductive bias across tasks is to design a parametrized hypothesis class that shares some parameters across tasks. One effective solution for MTL is finding solutions that are not dominated by any others, which is the same objective as MOO problems(Sener & Koltun, 2018).

### 5.1 EXPERIMENTS SETUP

**1. Datasets and Models.** 1). MultiMNIST Datasets and Learning Tasks: We test the convergence performance of MLM-CMOO using the "MultiMNIST" dataset (Sabour et al., 2017), which is a multi-task learning version of the MNIST dataset (LeCun et al., 1998) from LIBSVM repository. Specifically, to convert the hand-written classification problem into a multi-task problem, we randomly chose 60000 images. Images are divided into 2 tasks, and each task has $m = 30000$ samples. In our experiments, a group of images is positioned in the top left corner, while another group of images is positioned in the bottom right. The two tasks are task "L" (to categorize the top-left digit) and task "R" (to classify the bottom-right digit). The overall problem is to classify the images of different tasks. All algorithms use the same randomly generated initial point. The learning rates are chosen as $\eta = \beta = \alpha = 0.01$. we directly apply existing single-task MNIST models.

2). CelebA Dataset and Learning Tasks: We utilize the CelebA dataset (Liu et al., 2015), consisting of $200K$ facial images annotated with $40$ attributes. We approach each attribute as a binary classification task, resulting in a 40-way multi-task learning (MTL) problem. To create a shared representation function, we implement ResNet-18 (He et al., 2016) without the final layer, attaching a linear layer to each attribute for classification. In this experiment, we set $\eta = 0.0005, \alpha = \beta = 0.1$.

3). River Flow Dataset and Learning Tasks: We further test our algorithms on MOO problems of larger sizes. In this experiment, we use the River Flow dataset (Nie et al., 2017), which is for flow prediction flow at eight locations within the Mississippi River network. Thus, there are eight tasks in this problem. In this experiment, we set $\eta = 0.001, \alpha = \beta = 0.1$. To better visualize 8 different tasks, we illustrate the normalized loss in radar charts.

**2. Baseline.** The NSGA-II (Deb et al., 2002) and PSL (Lin et al., 2022) are considered as our baselines. NSGA-II is a well-known MOO evolutionary algorithm, while PSL is a novel MOO Bayesian optimization algorithm. NSGA-II and PSL do not handle constraints. For a fair comparison, we report the Pareto optimal solutions satisfying constraints generated from NSGA-II and PSL, thus NSGA-II and PSL can work towards the Pareto optimal solutions without considering constraints.

**NSGA-II Setup**. For binary chromosomes, we apply a single-point crossover with a probability of 0.9 and a bit-flip mutation with a probability of 0.1. For real-valued chromosomes, we apply a simulated binary crossover (SBX) [14] with a probability of 0.9 and $n_c = 2$ and a polynomial mutation with a probability of 0.1 and $n_m = 20$, where $n_c$ and $n_m$ denote spread factor distribution indices for crossover and mutation, respectively.

**PSL Setup**. We follow literature (Lin et al., 2022) to set PSL parameters. At each iteration, we train the Pareto set model $h_\theta$ with 1000 update steps using Adam optimizer with a learning rate of $1e^{-5}$ and no weight decay. At each iteration, we generate 1000 candidate solutions using $h_\theta$ and select the population size of solutions from the 1000 candidates.

## 5.2 EXPERIMENTS RESULTS

For the MultiMNIST Datasets and Learning Tasks. Table 5.2 shows that the MLM-CMOO outperforms than the rest 2 baselines. This is because that NSGA-II and PSL are designed for unconstrained MOO problems, even we have generated the Pareto optimal solutions satisfying constraints from those algorithm, their evolutionary or Bayesian Optimisation are not the most fit algorithm. On the contrary, MLM-CMOO is specifically designed for constrained MOO problems, where it uses an increasing Lagrange coefficient ($\mu$) to make it follow the limits. In this way, MLM-CMOO is more time-efficient. The Fig 1(a) and Fig 1(b) shows the results of 3 algorithm on CelebA Dataset and Learning Tasks. It shows that MLM-CMOO matches the selected baselines. The Fig 1(c) shows that MLM-CMOO's loss on River Flow Dataset and Learning Tasks is better than selected algoritms.

Table 1: Completion time are taken to reach Pareto stationary point with a specified loss for different algorithms using MultiMNIST Datasets.

| Loss | $10^{-2}$ | | | | | |
|------|-----------|--|--|--|--|--|
| Task | Task L | | | Task R | | |
| Algorithm | NSGA-II | PSL | MLM-CMOO | NSGA-II | PSL | MLM-CMOO |
| Time (s) | $\times$ 2.2 | $\times$ 1.8 | 1302 | $\times$ 2.3 | $\times$ 1.7 | 1322 |
| Loss | $10^{-3}$ | | | | | |
| Task | Task L | | | Task R | | |
| Algorithm | NSGA-II | PSL | MLM-CMOO | NSGA-II | PSL | MLM-CMOO |
| Time (s) | $\times$ 3.2 | $\times$ 2.03 | 2057 | $\times$ 3.1 | $\times$ 2.1 | 2104 |

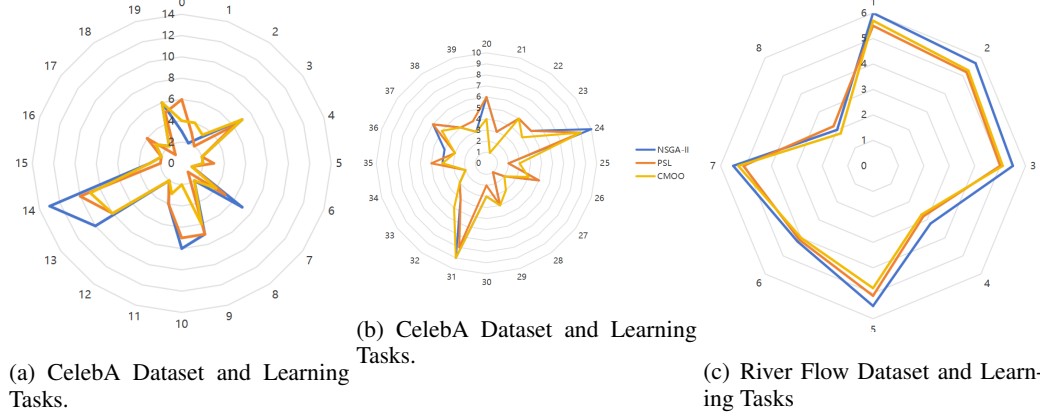

(a) CelebA Dataset and Learning Tasks.

(b) CelebA Dataset and Learning Tasks.

(c) River Flow Dataset and Learning Tasks

Figure 1: Experiments on CelebA dataset and River Flow Dataset.

## 6 CONCLUSION

This paper studies the constrained multi-objective optimization problem. We first establish a framework for the CMOO problem, which is suitable for gradient descent algorithms. For the formulated CMOO problem, we use the Lagrange Multiplier method to make a decrease in the overall objective obey the constraints. Then, due to the non-smoothness of the coefficient function, we use a Moreau envelope to make it smooth. Next, the convergence analysis shows that the proposed algorithm (MLM-CMOO) convergence to Pareto stationary solutions with a rate of $\mathcal{O}(\frac{1}{\sqrt{T}})$. Finally, we conduct experiments to verify the effectiveness of the MLM-CMOO algorithm.

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

## Supplementary Material for "Constrained Multi-Objective Optimization"

## A   SOME USEFUL LEMMA

To simplify the notions, we use $\boldsymbol{\omega}^{(t)} = (x^{(t)}, \boldsymbol{\lambda}^{(t)}, \boldsymbol{z}^{(t)})$ in the following proof.

**Lemma A.1** *Yao et al. (2024)* ***Lemma A.1****.   Under Assumption 4.3 and 4.4, for the Moreau envelope-based Lagrange Multiplier function $L_s(x, \boldsymbol{\lambda}, \boldsymbol{z}$ with $\gamma_1 \in (0, 1/\rho_f)$ and $\gamma_2 > 0$. That is,*

*(1) The function $L_s(x, \boldsymbol{\lambda}, \boldsymbol{z})$ is continuously differentiable;*

*(2) The gradient of $L_s(x, \boldsymbol{\lambda}, \boldsymbol{z})$ has closed-form given by*

$$\nabla_x L_s(x, \boldsymbol{\lambda}, \boldsymbol{z}) = \arg\min_u \{H(u, \boldsymbol{\theta}^*) + \frac{1}{2}\|u - x\|^2\} + \sum_{i=1}^n \mu_i^* \nabla g_i(x),$$

$$\nabla_{\boldsymbol{\lambda}} L_s(x, \boldsymbol{\lambda}, \boldsymbol{z}) = \frac{\boldsymbol{\lambda} - \boldsymbol{\theta}^*}{\gamma_1},$$

$$\nabla_{\boldsymbol{\lambda}} L_s(x, \boldsymbol{\lambda}, \boldsymbol{z}) = \frac{\boldsymbol{\mu}^* - \boldsymbol{z}}{\gamma_2},$$

*where $\boldsymbol{\theta}^* := \boldsymbol{\theta}^*(x, \boldsymbol{\lambda}, \boldsymbol{z})$ and $\boldsymbol{\mu}^* := \boldsymbol{\mu}^*(x, \boldsymbol{\lambda}, \boldsymbol{z})$ is the unique saddle point of the following min-max problem:*

$$\min_{\boldsymbol{\theta}} \max_{\boldsymbol{\mu}} \left\{ H(x, \boldsymbol{\theta}) + \sum_{i=1}^N \mu_i g_i(x) + \frac{1}{2\gamma_1} \sum_{i=1}^N \|\theta_i - \lambda_i\|^2 - \frac{1}{2\gamma_2} \sum_{i=1}^N \|z_i - \mu_i\|^2 \right\}.$$

*(3) Furthermore, for any $\rho_v \geq \rho_f/(1 - \gamma_1 \rho_f)$, $L_s(x, \boldsymbol{\lambda}, \boldsymbol{z})$ is $\rho_v$-weakly convex with respect to variables $(x, \boldsymbol{\lambda})$ on for any fixed $\boldsymbol{z}$.*

*Proof:* The proof is similar to the proof of **Lemma A.1** in Yao et al. (2024).

**Lemma A.2** *Yao et al. (2024)* ***Lemma A.2*** *and* ***Lemma A.4****.*

*Under Assumption 4.3 and 4.4, let $\gamma_1 \in (0, 1/\rho_f)$ and $\gamma_2 > 0$. Then, for any $\rho_v \geq \rho_f/(1 - \gamma_1 \rho_f)$, the following inequality holds:*

$$-L_s(x_1, \boldsymbol{\lambda}, \boldsymbol{z}) \leq -L_s(x_2, \boldsymbol{\lambda}, \boldsymbol{z}) - \langle \nabla_x L_s(x_2, \boldsymbol{\lambda}, \boldsymbol{z}), x_1 - x_2 \rangle + \frac{\rho_v}{2}\|x_1 - x_2\|^2,$$

$$-L_s(x, \boldsymbol{\lambda}_1, \boldsymbol{z}) \leq -L_s(x, \boldsymbol{\lambda}_2, \boldsymbol{z}) - \langle \nabla_{\boldsymbol{\lambda}} L_s(x, \boldsymbol{\lambda}_2, \boldsymbol{z}), \boldsymbol{\lambda}_1 - \boldsymbol{\lambda}_2 \rangle + \frac{\rho_v}{2}\|\boldsymbol{\lambda}_1 - \boldsymbol{\lambda}_2\|^2,$$

$$-L_s(x, \boldsymbol{\lambda}, \boldsymbol{z}_1) \leq -L_s(x, \boldsymbol{\lambda}, \boldsymbol{z}_2) - \langle \nabla_{\boldsymbol{\lambda}} L_s(x, \boldsymbol{\lambda}x, \boldsymbol{z}_2), \boldsymbol{z}_1 - \boldsymbol{z}_2 \rangle + \frac{L_z}{2}\|\boldsymbol{z}_1 - \boldsymbol{z}_2\|^2,$$

*where $L_z := (\gamma_2 \rho_T + 1)/(\gamma_2^2 \rho_T)$.*

*Proof:* The first 2 conclusions follow directly from *Lemma A.2* that $L_s(x, \boldsymbol{\lambda}, \boldsymbol{z})$ is $\rho_v$-weakly convex with respect to variables $(x, \boldsymbol{\lambda})$ on for any fixed $\boldsymbol{z}$, and the third conclusion is similar to the proof of **Lemma A.4** in Yao et al. (2024).

**Lemma A.3** *Yao et al. (2024)* ***Lemma A.3****. Under Assumption 4.3 and 4.4, let $\gamma_1 \in (0, 1/\rho_f)$ and $\gamma_2 > 0$. Then, for any $(x_1, \boldsymbol{\lambda}_1, \boldsymbol{z}_1)$ and $(x_2, \boldsymbol{\lambda}_2, \boldsymbol{z}_2)$, the following Lipschitz property holds:*

$$\|(\boldsymbol{\theta}^*(x_1, \boldsymbol{\lambda}_1, \boldsymbol{z}_1), \boldsymbol{\mu}^*(x_1, \boldsymbol{\lambda}_1, \boldsymbol{z}_1)) - (\boldsymbol{\theta}^*(x_2, \boldsymbol{\lambda}_2, \boldsymbol{z}_2), \boldsymbol{\mu}^*(x_2, \boldsymbol{\lambda}_2, \boldsymbol{z}_2))\|$$

$$\leq \frac{L_f + L_g + C_Z L_g}{\rho_T} \|x_1 - x_2\| + \frac{1}{\gamma_1 \rho_T} \|\boldsymbol{\lambda}_1 - \boldsymbol{\lambda}_2\| + \frac{1}{\gamma_2 \rho_T} \|\boldsymbol{z}_1 - \boldsymbol{z}_2\|$$

$$\leq L_{\theta, \mu} \|(x_1, \boldsymbol{\lambda}_1, \boldsymbol{z}_1) - (x_2, \boldsymbol{\lambda}_2, \boldsymbol{z}_2)\|,$$

*where $\rho_T := \min\{1/\gamma_1 - \rho_f, 1/\gamma_2\}$, $C_Z = \max_{z \in Z} \|z\|$, and $L_{\theta, \mu} := \sqrt{3} \max\{L_f + L_g + C_Z L_g, 1/\gamma_1, 1/\gamma_2\}/\rho_T$.*

*Proof:* The proof is similar to the proof of **Lemma A.3** in Yao et al. (2024).

## B  PROOF OF MAIN THEOREM AND LEMMAS

### B.1  PROOF OF THEOREM 4.6

**Theorem 4.6** If Assumptions of Assumptions 4.2, 4.3 and 4.4 hold, let $\gamma_1 \in (0, 1/\rho_\gamma)$, $\gamma_2 > 0$, $c_t = \underline{c}(t+1)^p$ with $p \in (0, 1/2)$ and $\underline{c} > 0$. Pick $\eta_t \in (0, \rho_\gamma/L_B^2)$, then there exists $c_\alpha, c_\beta > 0$ such that when $\alpha \in (\underline{\alpha}, c_\alpha)$ and $\beta \in (\underline{\beta}, c_\beta)$, with $\underline{\alpha}, \underline{\beta} > 0$, the sequence of $(x^{(t)}, \boldsymbol{\lambda}^{(t)}, \boldsymbol{z}^{(t)}, \boldsymbol{\theta}^{(t)}, \boldsymbol{\mu}^{(t)})$ generated by Algorithm 1: MLM-CMOO satisfies

$$\min_t \left\| (\boldsymbol{\theta}^t, \boldsymbol{\mu}^t) - (\boldsymbol{\theta}_r^*(x^{(t)}, \boldsymbol{\lambda}^{(t)}, \boldsymbol{z}^{(t)}), \boldsymbol{\mu}_r^*(x^{(t)}, \boldsymbol{\lambda}^{(t)}, \boldsymbol{z}^{(t)})) \right\| = \mathcal{O}(\frac{1}{\sqrt{T}}),$$

and

$$\min_t R_t(x^{(t)}, \boldsymbol{\lambda}^{(t)}, \boldsymbol{z}^{(t)}) = \mathcal{O}(\frac{1}{\sqrt{T^{1-2p}}}).$$

*Proof:* First, using the descent lemma in *Lemma 4.5* and its condition, telescoping the inequality for $t = 0, 1, ..., T-1$, we get

$$V_T - V_0 \leq -\frac{1}{4\alpha} \sum_{t=0}^{T-1} \left( \left\| x^{(t+1)} - x^{(t)} \right\|^2 + \left\| \boldsymbol{\lambda}^{(t+1)} - \boldsymbol{\lambda}^{(t)} \right\|^2 \right) - \frac{1}{4\beta} \sum_{t=0}^{T-1} \left\| \boldsymbol{z}^{(t+1)} - \boldsymbol{z}^{(t)} \right\|^2$$

$$- \eta \rho_T C_{\theta, \mu} \sum_{t=0}^{T-1} \left\| (\boldsymbol{\theta}^{(t)}, \boldsymbol{\mu}^{(t)}) - (\boldsymbol{\theta}^*(x^{(t)}, \boldsymbol{\lambda}^{(t)}, \boldsymbol{z}^{(t)}), \boldsymbol{\mu}^*(x^{(t)}, \boldsymbol{\lambda}^{(t)}, \boldsymbol{z}^{(t)})) \right\|^2.$$

From assumptions, we have $\sum_{t=0}^{T-1} \left\| (\boldsymbol{\theta}^{(t)}, \boldsymbol{\mu}^{(t)}) - (\boldsymbol{\theta}^*(x^{(t)}, \boldsymbol{\lambda}^{(t)}, \boldsymbol{z}^{(t)}), \boldsymbol{\mu}^*(x^{(t)}, \boldsymbol{\lambda}^{(t)}, \boldsymbol{z}^{(t)})) \right\|^2$ is upper bounded, which is

$$\sum_{t=0}^{T-1} \left\| (\boldsymbol{\theta}^{(t)}, \boldsymbol{\mu}^{(t)}) - (\boldsymbol{\theta}^*(x^{(t)}, \boldsymbol{\lambda}^{(t)}, \boldsymbol{z}^{(t)}), \boldsymbol{\mu}^*(x^{(t)}, \boldsymbol{\lambda}^{(t)}, \boldsymbol{z}^{(t)})) \right\|^2 \leq +\infty.$$

Thus, we have

$$\min_t \left\| (\boldsymbol{\theta}^{(t)}, \boldsymbol{\mu}^{(t)}) - (\boldsymbol{\theta}^*(x^{(t)}, \boldsymbol{\lambda}^{(t)}, \boldsymbol{z}^{(t)}), \boldsymbol{\mu}^*(x^{(t)}, \boldsymbol{\lambda}^{(t)}, \boldsymbol{z}^{(t)})) \right\| = \mathcal{O}(\frac{1}{\sqrt{T}}).$$

Secondly, according to the update rule of variables $(x, y, z)$, we have

$$0 \in c_t(d_x^{(t)}, d_{\boldsymbol{\lambda}}^{(t)}) + \mathcal{M}_C(x^{(t)}, \boldsymbol{\lambda}^{(t)}) + \frac{c_t}{\alpha}((x^{(t+1)}, \boldsymbol{\lambda}^{(t+1)}) - (x^{(t)}, \boldsymbol{\lambda}^{(t)})),$$

$$0 \in c_t d_{\boldsymbol{z}}^{(t)} + \mathcal{M}_Z(\boldsymbol{z}^{(t+1)}) + \frac{c_t}{\beta}(\boldsymbol{z}^{(t+1)} - \boldsymbol{z}^{(t)}).$$

where $d_x^{(t)} = \frac{1}{c^{(t)}} \nabla_x F(x^{(t)}, \boldsymbol{\lambda}^{(t)}) + U + \sum_{i=1}^n \mu_i^{(t+1)} \nabla_x g_i(x^{(t)})$, $d_{\boldsymbol{\lambda}}^{(t)} = \frac{1}{c^{(t)}}(\boldsymbol{\lambda}^{(t)} - \boldsymbol{\lambda}') + V - \frac{1}{\gamma_1}(\boldsymbol{\lambda})^{(t)} - \boldsymbol{\theta})^{(t+1)})$, and $d_{\boldsymbol{z}}^{(t)} = \boldsymbol{\mu}^{(t+1)} - \boldsymbol{z}^{(t)}$. Note, $U = \arg\min_u \{H(u, \boldsymbol{\lambda}^{(t)}) + \frac{1}{2} \left\| u - x^{(t)} \right\|^2\} - \arg\min_u \{H(u, \boldsymbol{\theta}^{(t+1)}) + \frac{1}{2} \left\| u - x^{(t)} \right\|^2\}$, $\boldsymbol{\lambda}' = \arg\min_{\boldsymbol{\lambda}} \left\| \sum_{i=1}^m \lambda_i \nabla f_i(x^{(t)}) \right\|$, and $V = \arg\min_v \{H(x^{(t)}, v) + \frac{1}{2} \left\| v - \boldsymbol{\lambda}^{(t)} \right\|^2\} - \arg\min_v \{H(u, \boldsymbol{\theta}^{(t+1)}) + \frac{1}{2} \left\| u - x^{(t)} \right\|^2\}$.

By the meanings of $d_x^{(t)}$, $d_{\boldsymbol{\lambda}}^{(t)}$, and $d_{\boldsymbol{z}}^{(t)}$, we obtain

$$(e_{x,\boldsymbol{\lambda}}^{(t)}, e_{\boldsymbol{z}}^{(t)}) \in (\nabla F(x^{(t+1)}, \boldsymbol{\lambda}^{(t+1)}), 0) + c_t(\sum_{i=1}^n \mu_i \nabla g_i(x^{(t+1)}), 0)$$

$$- c_t(\nabla L_{i,s,r}(x^{(t+1)}, \boldsymbol{\lambda}^{(t+1)}, \boldsymbol{z}^{(t+1)}) + \mathcal{M}_{C \times Z}(x^{(t+1)}, \boldsymbol{\lambda}^{(t+1)}, \boldsymbol{z}^{(t+1)}),$$

where

$$e_{x,\boldsymbol{\lambda}}^{(t)} := \nabla_{x,\boldsymbol{\lambda}} \phi_{c_t}(x^{(t)}, \boldsymbol{\lambda}^{(t)}, \boldsymbol{z}^{(t)}) - c_t(d_x^{(t)}, d_{\boldsymbol{\lambda}}^{(t)}) - \frac{c_t}{\alpha} - ((x^{(t+1)}, \boldsymbol{\lambda}^{(t+1)}) - ((x^{(t)}, \boldsymbol{\lambda}^{(t)})),$$

$$e_{\boldsymbol{z}}^{(t)} := \nabla_{\boldsymbol{z}} \phi_{c_t}(x^{(t)}, \boldsymbol{\lambda}^{(t)}, \boldsymbol{z}^{(t)}) - c_t(d_x^{(t)}, d_{\boldsymbol{\lambda}}^{(t)}) - c_t d_{\boldsymbol{z}}^{(t)} - \frac{c_t}{\beta} - (\boldsymbol{z}^{(t+1)} - \boldsymbol{z}^{(t)})).$$

Next, we estimate $\left\| e_{x,\boldsymbol{\lambda}}^{(t)} \right\|$. Using the estimates in Yao et al. (2024), we have

$$\begin{aligned}
\left\| e_{x,\boldsymbol{\lambda}}^{(t)} \right\| \leq & c_t L_{\phi_1} \left\| (x^{(t+1)}, \boldsymbol{\lambda}^{(t+1)}, \boldsymbol{z}^{(t+1)}) - (x^{(t)}, \boldsymbol{\lambda}^{(t)}, \boldsymbol{z}^{(t)}) \right\| \\
& + \frac{c_t}{\alpha} \left\| (x^{(t+1)}, \boldsymbol{\lambda}^{(t+1)}) - (x^{(t)}, \boldsymbol{\lambda}^{(t)}) \right\| + c_t C_{\phi_1} \\
& + \left\| (\boldsymbol{\theta}^{(t)}, \boldsymbol{\mu}^{(t)}) - (\boldsymbol{\theta}^*(x^{(t)}, \boldsymbol{\lambda}^{(t)}, \boldsymbol{z}^{(t)}), \boldsymbol{\mu}^*(x^{(t)}, \boldsymbol{\lambda}^{(t)}, \boldsymbol{z}^{(t)})) \right\|,
\end{aligned}$$

where $C_{\phi_1} := \sqrt{\max\{2(L_g + C_z L_g)^2, 2L_g^2\}}$.

For $\left\| e_{\boldsymbol{z}}^{(t)} \right\|$, we have

$$\left\| e_{\boldsymbol{z}}^{(t)} \right\| \leq (\frac{c_t}{\beta} + \frac{c_t}{\gamma_2}) \left\| \boldsymbol{z}^{(t+1)} - \boldsymbol{z}^{(t)} \right\| + \frac{c_t}{\gamma_2} \left\| \boldsymbol{\mu}^{(t)} - \boldsymbol{\mu}^*(x^{(t)}, \boldsymbol{\lambda}^{(t)}, \boldsymbol{z}^{(t)}) \right\|.$$

Thus,

$$\begin{aligned}
R_t(x^{(t)}, \boldsymbol{\lambda}^{(t)}, \boldsymbol{z}^{(t)}) \leq & (\frac{c_t}{\beta} + \frac{c_t}{\gamma_2}) \left\| \boldsymbol{z}^{(t+1)} - \boldsymbol{z}^{(t)} \right\| + \frac{c_t}{\alpha} \left\| (x^{(t+1)}, \boldsymbol{\lambda}^{(t+1)}) - (x^{(t)}, \boldsymbol{\lambda}^{(t)}) \right\| \\
& + c_t(C_{\phi_1} + \frac{1}{\gamma_2}) \left\| (\boldsymbol{\theta}^{(t)}, \boldsymbol{\mu}^{(t)}) - (\boldsymbol{\theta}^*(x^{(t)}, \boldsymbol{\lambda}^{(t)}, \boldsymbol{z}^{(t)}), \boldsymbol{\mu}^*(x^{(t)}, \boldsymbol{\lambda}^{(t)}, \boldsymbol{z}^{(t)})) \right\| \\
& + c_t L_{\phi_1} \left\| (x^{(t+1)}, \boldsymbol{\lambda}^{(t+1)}, \boldsymbol{z}^{(t+1)}) - (x^{(t)}, \boldsymbol{\lambda}^{(t)}, \boldsymbol{z}^{(t)}) \right\|.
\end{aligned}$$

Let $\alpha_t \geq \underline{\alpha}$ and $\beta_t \geq \underline{\beta}$ for some positive constants $\underline{\alpha}$ and $\underline{\beta}$, we can show that there exists $C_R > 0$ such that

$$\begin{aligned}
\frac{1}{c_t^2} R_t^2(x^{(t)}, \boldsymbol{\lambda}^{(t)}, \boldsymbol{z}^{(t)}) \leq & C_R \left( \frac{1}{4\underline{\alpha}} \left\| (x^{(t+1)}, \boldsymbol{\lambda}^{(t+1)}) - (x^{(t)}, \boldsymbol{\lambda}^{(t)}) \right\|^2 + \frac{1}{4\underline{\beta}} \left\| \boldsymbol{z}^{(t+1)} - \boldsymbol{z}^{(t)} \right\|^2 \right. \\
& \left. + \eta \rho_T C_{\boldsymbol{\theta},\boldsymbol{\mu}} \left\| (\boldsymbol{\theta}^{(t)}, \boldsymbol{\mu}^{(t)}) - (\boldsymbol{\theta}^*(x^{(t)}, \boldsymbol{\lambda}^{(t)}, \boldsymbol{z}^{(t)}), \boldsymbol{\mu}^*(x^{(t)}, \boldsymbol{\lambda}^{(t)}, \boldsymbol{z}^{(t)})) \right\|^2 \right).
\end{aligned}$$

This completes the proof.

### B.2 PROOF OF LEMMA 4.7

**Lemma 4.7.** Under Assumption 4.3 and 4.4, let $\gamma_1 \in (0, 1/\rho_f)$, $\gamma_2 > 0$ and pick $\eta \in (0, \rho_T/L_b^2$, where $L_b := \max\{L_f + L_g + C_Z L_g + 1/\gamma_1, L_g + 1/\gamma_2\}$. Then, the sequence generated by Algorithm 1 satisfies

$$\begin{aligned}
& \left\| (\boldsymbol{\theta}^{(t+1)}, \boldsymbol{\mu}^{(t+1)}) - \left( \boldsymbol{\theta}^*(\boldsymbol{\omega}^{(t)}), \boldsymbol{\mu}^*(\boldsymbol{\omega}^{(t)}) \right) \right\| \\
& \leq (1 - \eta \rho_T) \left\| (\boldsymbol{\theta}^{(t)}, \boldsymbol{\mu}^{(t)}) - \left( \boldsymbol{\theta}^*(\boldsymbol{\omega}^{(t)}), \boldsymbol{\mu}^*(\boldsymbol{\omega}^{(t)}) \right) \right\|.
\end{aligned}$$

*Proof:* The proof is similar to the proof of **Lemma A.5** in Yao et al. (2024).

### B.3 PROOF OF LEMMA 4.8

**Lemma 4.8.** Suppose the assumption of 4.2, 4.3 and 4.4 hold, and let $\gamma_1 \in (0, 1/\rho_g)$, $\gamma_2 > 0$. Pick $\eta \in (0, \rho_\gamma/L_B^2)$ with $L_B := \max\{2L_g + C_z L_g + 1/\gamma_1, L_g + 1/\gamma_2\}$ then the sequence of $(\boldsymbol{\omega}^{(t)})$ generated by Algorithm 1: MLM-CMOO satisfies

$$\phi_{c_t}(\boldsymbol{\omega}^{(t+1)}) \leq \phi_{c_t}(\boldsymbol{\omega}^{(t)}) - (\frac{1}{2\beta} - \frac{L_{v_z}}{2}) \left\| \boldsymbol{z}^{(t+1)} - \boldsymbol{z}^{(t)} \right\|^2$$

$$- \left( \frac{1}{2\alpha} - \frac{L_{\phi_k}}{2} - \frac{\beta L_{\theta,\mu}^2}{\gamma_2^2} \right) \left( \left\| x^{(t+1)} - x^{(t)} \right\|^2 + \left\| \boldsymbol{\lambda}^{(t+1)} - \boldsymbol{\lambda}^{(t)} \right\|^2 \right)$$

$$+ \frac{\alpha}{2} \left( 2(L_g + C_z L_g)^2 + \frac{1}{\gamma_1^2} \right) \left\| \boldsymbol{\theta}^{(t+1)} - \boldsymbol{\theta}^*(\boldsymbol{\omega}^{(t)}) \right\|^2$$

$$+ (\alpha L_g^2 + \frac{\beta}{\gamma_2^2}) \left\| \boldsymbol{\mu}^{(t+1)} - \boldsymbol{\mu}^*(\boldsymbol{\omega}^{(t)}) \right\|^2,$$

where $L_{\phi_t} := L_f/c_t + L_g + \rho_v$.

*Proof:* Given Assumptions 4.2, 4.3, and 4.4 that $\nabla F$ and $\nabla g$ are $L_F$- and $L_g$-Lipschitz continuous on their domain, respectively, and applying **Lemma 5.7** in Beck (2017)] and previous Lemmas, we obtain

$$\phi_{c_t}(\boldsymbol{\omega}^{(t+1)}) \leq \phi_{c_t}(\boldsymbol{\omega}^{(t)}) + \left\langle \nabla_{x,\boldsymbol{\lambda}} \phi_{c_t}(\boldsymbol{\omega}^{(t)}), (x^{(t+1)}, \boldsymbol{\lambda}^{(t+1)}) - (x^{(t)}, \boldsymbol{\lambda}^{(t)}) \right\rangle$$

$$+ \frac{L_{\phi_t}}{2} \left\| (x^{(t+1)}, \boldsymbol{\lambda}^{(t+1)}) - (x^{(t)}, \boldsymbol{\lambda}^{(t)}) \right\|^2,$$

with $L_{\phi_t} := L_F/c_t + L_g + \rho_v$. Based on the update rule of variable $x^{(t)}, \boldsymbol{\lambda}^{(t)}$, the convexity and the property of the proximal operator, we have

$$\left\langle (x^{(t)}, \boldsymbol{\lambda}^{(t)}) - \alpha(d_x^{(t)}, d_{\boldsymbol{\lambda}}^{(t)}) - (x^{(t+1)}, \boldsymbol{\lambda}^{(t+1)}), (x^{(t)}, \boldsymbol{\lambda}^{(t)}) - (x^{(t+1)}, \boldsymbol{\lambda}^{(t+1)}) \right\rangle \leq 0,$$

thus, we have

$$\left\langle (d_x^{(t)}, d_{\boldsymbol{\lambda}}^{(t)}), (x^{(t+1)}, \boldsymbol{\lambda}^{(t+1)}) - (x^{(t)}, \boldsymbol{\lambda}^{(t)}) \right\rangle \leq -\frac{1}{\alpha} \left\| (x^{(t+1)}, \boldsymbol{\lambda}^{(t+1)}) - (x^{(t)}, \boldsymbol{\lambda}^{(t)}) \right\|^2.$$

Considering the formula of $\nabla_{x,\boldsymbol{\lambda}} L_{s,r}$ derived in **Lemma A.2** and the meanings of $d_x^{(t)}$, $d_{\boldsymbol{\lambda}}^{(t)}$ provided in the previous proof, we can obtain that

$$\left\| \nabla_{x,\boldsymbol{\lambda}} L_{s,r}(\boldsymbol{\omega}^{(t)}) - (d_x^{(t)}, d_{\boldsymbol{\lambda}}^{(t)}) \right\|^2$$

$$= \left\| \nabla_x H(x^{(t)}, \boldsymbol{\theta}^*(\boldsymbol{\omega}^{(t)})) + \sum_{i=1}^n \mu_i^*(\boldsymbol{\omega}^{(t)}) \nabla_x g(x^{(t)}) - \nabla_x H(x^{(t)}, \boldsymbol{\theta}^{(t+1)}) - \sum_{i=1}^n \mu_i^{(t+1)} \nabla_x g(x^{(t)}) \right\|^2$$

$$+ \frac{1}{\gamma_1^2} \left\| \boldsymbol{\theta}^{(t+1)} - \boldsymbol{\theta}^*(\boldsymbol{\omega}^{(t)}) \right\|^2$$

$$\leq 2 \left\| \nabla_x H(x^{(t)}, \boldsymbol{\theta}^*(\boldsymbol{\omega}^{(t)})) + \sum_{i=1}^n \mu_i^*(\boldsymbol{\omega}^{(t)}) \nabla_x g(x^{(t)}) - \nabla_x H(x^{(t)}, \boldsymbol{\theta}^*(\boldsymbol{\omega}^{(t)})) - \sum_{i=1}^n \mu_i^{(t+1)} \nabla_x g(x^{(t)}) \right\|^2$$

$$+ 2 \left\| \nabla_x H(x^{(t)}, \boldsymbol{\theta}^*(\boldsymbol{\omega}^{(t)})) + \sum_{i=1}^n \mu_i^{(t+1)} \nabla_x g(x^{(t)}) - \nabla_x H(x^{(t)}, \boldsymbol{\theta}^{(t+1)}) - \sum_{i=1}^n \mu_i^{(t+1)} \nabla_x g(x^{(t)}) \right\|^2$$

$$+ \frac{1}{\gamma_1^2} \left\| \boldsymbol{\theta}^{(t+1)} - \boldsymbol{\theta}^*(\boldsymbol{\omega}^{(t)}) \right\|^2$$

$$\leq \left( 2(L_f + C_Z L_g + \frac{1}{\gamma_1^2}) \right) \left\| \boldsymbol{\theta}^{(t+1)}) - \boldsymbol{\theta}^*(\boldsymbol{\omega}^{(t)}) \right\|^2 + 2L_g^2 \left\| \boldsymbol{\mu}^{(t+1)}) - \boldsymbol{\mu}^*(\boldsymbol{\omega}^{(t)}) \right\|^2,$$

which yields

$$\left\langle \nabla_{x,\boldsymbol{\lambda}} L_{s,r}(\boldsymbol{\omega}^{(t)}) - (d_x^{(t)}, d_{\boldsymbol{\lambda}}^{(t)}), \boldsymbol{\lambda}^{(t+1)}), (x^{(t+1)}, \boldsymbol{\lambda}^{(t+1)}) - (x^{(t)}, \boldsymbol{\lambda}^{(t)}) \right\rangle$$

$$\leq \frac{\alpha}{2} \left( 2(L_f + C_Z L_g + \frac{1}{\gamma_1^2}) \right) \left\| \boldsymbol{\theta}^{(t+1)}) - \boldsymbol{\theta}^*(\boldsymbol{\omega}^{(t)}) \right\|^2 + \alpha L_g^2 \left\| \boldsymbol{\mu}^{(t+1)}) - \boldsymbol{\mu}^*(\boldsymbol{\omega}^{(t)}) \right\|^2$$

$$+ \frac{1}{2\alpha} \left\| (x^{(t+1)}, \boldsymbol{\lambda}^{(t+1)}) - (x^{(t)}, \boldsymbol{\lambda}^{(t)}) \right\|^2,$$

Combing with the above inequalities, we have

$$
\begin{aligned}
\phi_{c_t}(\boldsymbol{\omega}^{(t+1)}) \leq & \phi_{c_t}(\boldsymbol{\omega}^{(t)}) + \left(\frac{1}{2\alpha} - \frac{L_{\phi_t}}{2}\right) \left\|(x^{(t+1)}, \boldsymbol{\lambda}^{(t+1)}) - (x^{(t)}, \boldsymbol{\lambda}^{(t)})\right\|^2 \\
& + \frac{\alpha}{2}\left(2(L_f + C_Z L_g + \frac{1}{\gamma_1^2})\left\|\boldsymbol{\theta}^{(t+1)}) - \boldsymbol{\theta}^*(\boldsymbol{\omega}^{(t)})\right\|^2 + \alpha L_g^2 \left\|\boldsymbol{\mu}^{(t+1)}) - \boldsymbol{\mu}^*(\boldsymbol{\omega}^{(t)})\right\|^2
\end{aligned}
$$

For variable $\boldsymbol{z}$, we have

$$
\phi_{c_t}(\boldsymbol{\omega}^{(t+1)}) \leq \phi_{c_t}(\boldsymbol{\omega}^{(t)}) + \left\langle \nabla_{\boldsymbol{z}}\phi_{c_t}(\boldsymbol{\omega}^{(t)}), \boldsymbol{z}^{(t+1)} - \boldsymbol{z}^{(t)} \right\rangle + \frac{L_z}{2}\left\|\boldsymbol{z}^{(t+1)} - \boldsymbol{z}^{(t)}\right\|^2.
$$

According to the property of the proximal gradient, we have

$$
\left\langle d_{\boldsymbol{z}}^{(t)}, \boldsymbol{z}^{(t+1)} - \boldsymbol{z}^{(t)} \right\rangle \leq -\frac{1}{\beta}\left\|\boldsymbol{z}^{(t+1)} - \boldsymbol{z}^{(t)}\right\|^2
$$

Thus, we have

$$
\phi_{c_t}(\boldsymbol{\omega}^{(t+1)}) \leq \phi_{c_t}(\boldsymbol{\omega}^{(t)}) + \left\langle \nabla_{\boldsymbol{z}}\phi_{c_t}(\boldsymbol{\omega}^{(t)}) - d_{\boldsymbol{z}}^{(t)}, \boldsymbol{z}^{(t+1)} - \boldsymbol{z}^{(t)} \right\rangle + (\frac{L_z}{2} - \frac{1}{\beta})\left\|\boldsymbol{z}^{(t+1)} - \boldsymbol{z}^{(t)}\right\|^2.
$$

Based on the definition of $d_{\boldsymbol{z}}^{(t)}$ provided in the previous section, we have

$$
\left\|\boldsymbol{\omega}^{(t)} - d_{\boldsymbol{z}}^{(t)}\right\|^2 \leq \frac{1}{\gamma_2^2}\left\|\boldsymbol{\mu}^{(t+1)} - \boldsymbol{\mu}^*(x^{(t+1)}, \boldsymbol{\lambda}^{(t+1)}, \boldsymbol{z}^{(t)})\right\|^2,
$$

and

$$
\left\langle \nabla_{\boldsymbol{z}}\phi_{c_t}(\boldsymbol{\omega}^{(t)}) - d_{\boldsymbol{z}}^{(t)}, \boldsymbol{z}^{(t+1)} - \boldsymbol{z}^{(t)} \right\rangle \leq \frac{\beta}{2\gamma_2^2}\left\|\boldsymbol{\mu}^{(t+1)} - \boldsymbol{\mu}^*(x^{(t+1)}, \boldsymbol{\lambda}^{(t+1)}, \boldsymbol{z}^{(t)})\right\|^2 + \frac{1}{2\beta}\left\|\boldsymbol{z}^{(t+1)} - \boldsymbol{z}^{(t)}\right\|^2
$$

The, for variable $\boldsymbol{z}$, we can get

$$
\begin{aligned}
\phi_{c_t}(\boldsymbol{\omega}^{(t+1)}) \leq & \phi_{c_t}(\boldsymbol{\omega}^{(t)}) + \frac{\beta}{2\gamma_2^2}\left\|\boldsymbol{\mu}^{(t+1)} - \boldsymbol{\mu}^*(x^{(t+1)}, \boldsymbol{\lambda}^{(t+1)}, \boldsymbol{z}^{(t)})\right\|^2 + (\frac{L_z}{2} - \frac{1}{2\beta})\left\|\boldsymbol{z}^{(t+1)} - \boldsymbol{z}^{(t)}\right\|^2 \\
\leq & \phi_{c_t}(\boldsymbol{\omega}^{(t)}) + \frac{\beta}{2\gamma_2^2}\left\|\boldsymbol{\mu}^{(t+1)} - \boldsymbol{\mu}^*(x^{(t)}, \boldsymbol{\lambda}^{(t)}, \boldsymbol{z}^{(t)})\right\|^2 + (\frac{L_z}{2} - \frac{1}{2\beta})\left\|\boldsymbol{z}^{(t+1)} - \boldsymbol{z}^{(t)}\right\|^2 \\
& + \frac{\beta L_{\boldsymbol{\theta},\boldsymbol{\mu}}^2}{2\gamma_2^2}\left\|(x^{(t+1)}, \boldsymbol{\lambda}^{(t+1)}) - (x^{(t)}, \boldsymbol{\lambda}^{(t)})\right\|^2.
\end{aligned}
$$

Combining the inequities for variable $(x, \boldsymbol{\lambda})$ and $\boldsymbol{z}$, we can get Lemma 4.8.

### B.4 PROOF OF LEMMA 4.5

**Lemma 4.5** Under Assumptions 4.2, 4.3 and 4.4 hold, let $\gamma_1 \in (0, 1/\rho_T)$, $\gamma_2 > 0$, $c_t \leq c_{t+1}$ and $\eta_t \in (\eta, \rho_\gamma/L_B^2)$ with $\eta > 0$, then there exist constants $c_\alpha, c_\beta > 0$ such that when $0 < \alpha \leq c_\alpha$ and $0 < \beta \leq c_\beta$, the sequence of $(x^{(t)}, \boldsymbol{\lambda}^{(t)}, \boldsymbol{z}^{(t)})$ generated by Algorithm 1: MLM-CMOO satisfies

$$
\begin{aligned}
V_{t+1} - V_t \leq & -\frac{1}{4\alpha}\left\|x^{(t+1)} - x^{(t)}\right\|^2 - \frac{1}{4\alpha}\left\|\boldsymbol{\lambda}^{(t+1)} - \boldsymbol{\lambda}^{(t)}\right\|^2 - \frac{1}{4\beta}\left\|\boldsymbol{z}^{(t+1)} - \boldsymbol{z}^{(t)}\right\|^2 \\
& - \eta\rho_T C_{\boldsymbol{\theta},\boldsymbol{\mu}}\left\|(\boldsymbol{\theta}^{(t)}, \boldsymbol{\mu}^{(t)}) - (\boldsymbol{\theta}^*(x^{(t)}, \boldsymbol{\lambda}^{(t)}, \boldsymbol{z}^{(t)}), \boldsymbol{\mu}^*(x^{(t)}, \boldsymbol{\lambda}^{(t)}, \boldsymbol{z}^{(t)}))\right\|^2.
\end{aligned}
$$

*Proof:* From Lemma 4.8 and server aggregation rule, we have

$$
\phi_{c_t}(\boldsymbol{\omega}^{(t)}) \leq \phi_{c_t}(\boldsymbol{\omega}^{(t)}) - (\frac{1}{2\beta} - \frac{L_{v_z}}{2})\left\|\boldsymbol{z}^{(t+1)} - \boldsymbol{z}^{(t)}\right\|^2 \tag{8}
$$

$$- \left( \frac{1}{2\alpha} - \frac{L_{\phi_k}}{2} - \frac{\beta L_{\theta,\mu}^2}{\gamma_2^2} \right) \left( \left\| x^{(t+1)} - x^{(t)} \right\|^2 + \left\| \boldsymbol{\lambda}^{(t+1)} - \boldsymbol{\lambda}^{(t)} \right\|^2 \right)$$

$$+ \frac{\alpha}{2} \left( 2(L_g + C_z L_g)^2 + \frac{1}{\gamma_1^2} \right) \left\| \boldsymbol{\theta}^{(t+1)} - \boldsymbol{\theta}^*(\boldsymbol{\omega}^{(t)}) \right\|^2$$

$$+ (\alpha L_g^2 + \frac{\beta}{\gamma_2^2}) \left\| \boldsymbol{\mu}^{(t+1)} - \boldsymbol{\mu}^*(\boldsymbol{\omega}^{(t)}) \right\|^2 .$$

Since $c_{t+1} \geq c_t$, we can infer that $(F(x^{(t)}, \boldsymbol{\lambda}^{(t)}) - \underline{F})/c_{t+1} \leq (F(x^{(t)}, \boldsymbol{\lambda}^{(t)}) - \underline{F})/c_t$. Combining with inequality equation 8 leads to

$$V_{t+1} - V_t = \phi_{c_{t+1}}(\boldsymbol{\omega}^{(t+1)})) - \phi_{c_t}(\boldsymbol{\omega}^{(t)})) + C_{\boldsymbol{\theta},\boldsymbol{\mu}} \left\| (\boldsymbol{\theta}^{(t+1)}, \boldsymbol{\mu}^{(t+1)}) - (\boldsymbol{\theta}^*(\boldsymbol{\omega}^{(t+1)}), \boldsymbol{\mu}^*(\boldsymbol{\omega}^{(t+1)})) \right\|^2$$

$$- C_{\boldsymbol{\theta},\boldsymbol{\mu}} \left\| (\boldsymbol{\theta}^{(t)}, \boldsymbol{\mu}^{(t)}) - (\boldsymbol{\theta}^*(\boldsymbol{\omega}^{(t)}), \mu^*(\boldsymbol{\omega}^{(t)})) \right\|^2$$

$$\leq - (\frac{1}{2\alpha} - \frac{L_{\phi_t}}{2} - \frac{\beta L_{\theta,\mu}^2}{\gamma_2^2}) \left\| (x^{(t+1)}, \boldsymbol{\lambda}^{(t+1)}) - (x^{(t)}, \boldsymbol{\lambda}^{(t)}) \right\|^2 - (\frac{1}{2\beta} - \frac{L_{v_z}}{2}) \left\| \boldsymbol{z}^{(t+1)} - \boldsymbol{z}^{(t)} \right\|^2$$

$$+ (\alpha L_g^2 + \frac{\beta}{\gamma_2^2}) \left\| \boldsymbol{\mu}^{(t+1)} - \boldsymbol{\mu}^*(\boldsymbol{\omega}^{(t)}) \right\|^2 + \frac{\alpha}{2} \left( 2(L_g + C_z L_g)^2 + \frac{1}{\gamma_1^2} \right) \left\| \boldsymbol{\theta}^{(t+1)} - \boldsymbol{\theta}^*(\boldsymbol{\omega}^{(t)}) \right\|^2$$

$$+ C_{\boldsymbol{\theta},\boldsymbol{\mu}} \left\| (\boldsymbol{\theta}^{(t+1)}, \boldsymbol{\mu}^{(t+1)}) - (\boldsymbol{\theta}^*(\boldsymbol{\omega}^{(t+1)}), \boldsymbol{\mu}^*(\boldsymbol{\omega}^{(t+1)})) \right\|^2$$

$$- C_{\boldsymbol{\theta},\boldsymbol{\mu}} \left\| (\boldsymbol{\theta}^{(t)}, \boldsymbol{\mu}^{(t)}) - (\boldsymbol{\theta}^*(\boldsymbol{\omega}^{(t)}), \mu^*(\boldsymbol{\omega}^{(t)})) \right\|^2$$

$$\leq - (\frac{1}{2\alpha} - \frac{L_{\phi_t}}{2} - \frac{\beta L_{\theta,\mu}^2}{\gamma_2^2}) \left\| (x^{(t+1)}, \boldsymbol{\lambda}^{(t+1)}) - (x^{(t)}, \boldsymbol{\lambda}^{(t)}) \right\|^2 - (\frac{1}{2\beta} - \frac{L_{v_z}}{2}) \left\| \boldsymbol{z}^{(t+1)} - \boldsymbol{z}^{(t)} \right\|^2$$

$$+ C_{\theta,\lambda} \left\{ - \left\| (\boldsymbol{\theta}^{(t)}, \boldsymbol{\mu}^{(t)}) - (\boldsymbol{\theta}^*(\boldsymbol{\omega}^{(t)}), \mu^*(\boldsymbol{\omega}^{(t)})) \right\|^2 + \left\| (\boldsymbol{\theta}^{(t)}, \boldsymbol{\mu}^{(t)}) - (\boldsymbol{\theta}^*(\boldsymbol{\omega}^{(t)}), \mu^*(\boldsymbol{\omega}^{(t)})) \right\|^2 \right.$$

$$\left. + 2 \max\{\alpha, \beta\} \left\| (\boldsymbol{\theta}^{(t+1)}, \boldsymbol{\mu}^{(t+1)}) - (\boldsymbol{\theta}^*(\boldsymbol{\omega}^{(t)}), \mu^*(\boldsymbol{\omega}^{(t)})) \right\|^2 \right\},$$

where the last inequality follows from the fact that $C_{\theta,\lambda} := \max\{(L_g + C_Z L_g)^2 + 1/(2\gamma_1^2) + L_g^2, 1/\gamma_2^2\}$.

Then, for the last 3 terms in the previous equation, we have

$$- \left\| (\boldsymbol{\theta}^{(t)}, \boldsymbol{\mu}^{(t)}) - (\boldsymbol{\theta}^*(\boldsymbol{\omega}^{(t)}), \mu^*(\boldsymbol{\omega}^{(t)})) \right\|^2 + \left\| (\boldsymbol{\theta}^{(t)}, \boldsymbol{\mu}^{(t)}) - (\boldsymbol{\theta}^*(\boldsymbol{\omega}^{(t)}), \mu^*(\boldsymbol{\omega}^{(t)})) \right\|^2$$

$$+ 2\alpha \left\| (\boldsymbol{\theta}^{(t+1)}, \boldsymbol{\mu}^{(t+1)}) - (\boldsymbol{\theta}^*(\boldsymbol{\omega}^{(t)}), \mu^*(\boldsymbol{\omega}^{(t)})) \right\|^2$$

$$\overset{a}{\leq} (1 + \frac{1}{\epsilon_t}) \left\| (\boldsymbol{\theta}^*(\boldsymbol{\omega}^{(t+1)}), \mu^*(\boldsymbol{\omega}^{(t+1)}) - (\boldsymbol{\theta}^*(\boldsymbol{\omega}^{(t)}), \mu^*(\boldsymbol{\omega}^{(t)}) \right\|^2$$

$$- \left\| (\boldsymbol{\theta}^{(t)}, \boldsymbol{\mu}^{(t)}) - (\boldsymbol{\theta}^*(\boldsymbol{\omega}^{(t)}), \mu^*(\boldsymbol{\omega}^{(t)})) \right\|^2$$

$$+ (1 + \epsilon_t + 2\alpha) \left\| (\boldsymbol{\theta}^{(t+1)}, \boldsymbol{\mu}^{(t+1)}) - (\boldsymbol{\theta}^*(\boldsymbol{\omega}^{(t)}), \mu^*(\boldsymbol{\omega}^{(t)})) \right\|^2$$

$$\overset{b}{\leq} (1 + \frac{1}{\epsilon_t}) L_{\boldsymbol{\theta},\boldsymbol{\mu}} \left\| \boldsymbol{\omega}^{(t+1)} - \boldsymbol{\omega}^{(t)} \right\|^2 - \left\| (\boldsymbol{\theta}^{(t)}, \boldsymbol{\mu}^{(t)}) - (\boldsymbol{\theta}^*(\boldsymbol{\omega}^{(t)}), \mu^*(\boldsymbol{\omega}^{(t)})) \right\|^2$$

$$+ (1 + \epsilon_t + 2\alpha)(1 - \eta\rho_T)^2 \left\| (\boldsymbol{\theta}^{(t+1)}, \boldsymbol{\mu}^{(t+1)}) - (\boldsymbol{\theta}^*(\boldsymbol{\omega}^{(t)}), \mu^*(\boldsymbol{\omega}^{(t)})) \right\|^2$$

$$\leq (1 + \frac{2}{\eta\rho_T}) L_{\boldsymbol{\theta},\boldsymbol{\mu}}^2 \left\| \boldsymbol{\omega}^{(t+1)} - \boldsymbol{\omega}^{(t)} \right\|^2 - \eta\rho_T \left\| (\boldsymbol{\theta}^{(t)}, \boldsymbol{\mu}^{(t)}) - (\boldsymbol{\theta}^*(\boldsymbol{\omega}^{(t)}), \mu^*(\boldsymbol{\omega}^{(t)})) \right\|^2 ,$$

where $a$ from Lemma A.5 and A.7 for $\epsilon > 0$, and $b$ from setting $\epsilon = \eta\rho_T/2$ and picking $\alpha \leq \eta\rho_T/4$ where holds that $(1 + \epsilon + 2\alpha)(1 - \eta\rho_T) \leq 1$.

Similarly, we can show that when $\beta \leq \eta \rho_T / 4$, it holds that

$$- \left\| (\boldsymbol{\theta}^{(t)}, \boldsymbol{\mu}^{(t)}) - (\boldsymbol{\theta}^*(\boldsymbol{\omega}^{(t)}), \mu^*(\boldsymbol{\omega}^{(t)})) \right\|^2 + \left\| (\boldsymbol{\theta}^{(t)}, \boldsymbol{\mu}^{(t)}) - (\boldsymbol{\theta}^*(\boldsymbol{\omega}^{(t)}), \mu^*(\boldsymbol{\omega}^{(t)})) \right\|^2$$

$$\leq (1 + \frac{2}{\eta \rho_T}) L_{\boldsymbol{\theta}, \boldsymbol{\mu}}^2 \left\| \boldsymbol{\omega}^{(t+1)} - \boldsymbol{\omega}^{(t)} \right\|^2 - \eta \rho_T \left\| (\boldsymbol{\theta}^{(t)}, \boldsymbol{\mu}^{(t)}) - (\boldsymbol{\theta}^*(\boldsymbol{\omega}^{(t)}), \mu^*(\boldsymbol{\omega}^{(t)})) \right\|^2.$$

Combining the above inequities, we have

$$V_{t+1} - V_t \leq - \left( \frac{1}{2\alpha} - \frac{L_{\phi_t}}{2} - \frac{\beta L_{\boldsymbol{\theta}, \boldsymbol{\mu}}^2}{\gamma_2^2} - (1 + \frac{2}{\eta \rho_T}) L_{\boldsymbol{\theta}, \boldsymbol{\mu}}^2 C_{\theta, \lambda} \right) \left\| (x^{(t+1)}, \boldsymbol{\lambda}^{(t+1)}) - (x^{(t)}, \boldsymbol{\lambda}^{(t)}) \right\|^2$$

$$- \left( \frac{1}{2\beta} - \frac{L_{v_z}}{2} - (1 + \frac{2}{\eta \rho_T}) L_{\boldsymbol{\theta}, \boldsymbol{\mu}}^2 C_{\theta, \lambda} \right) \left\| \boldsymbol{z}^{(t+1)} - \boldsymbol{z}^{(t)} \right\|^2$$

$$+ \eta \rho_T C_{\theta, \lambda} \left\| (\boldsymbol{\theta}^{(t)}, \boldsymbol{\mu}^{(t)}) - (\boldsymbol{\theta}^*(\boldsymbol{\omega}^{(t)}), \mu^*(\boldsymbol{\omega}^{(t)})) \right\|^2.$$

When $c_{t+1} \geq c_t, \eta \geq \underline{\eta} > 0, \alpha \leq \underline{\eta} \rho_T / 4$ and $\beta \leq \underline{\eta} \rho_T / 4$, then $\frac{L_{\phi_t}}{2} + \frac{\beta L_{\boldsymbol{\theta}, \boldsymbol{\mu}}^2}{\gamma_2^2} + (1 + \frac{2}{\eta \rho_T}) L_{\boldsymbol{\theta}, \boldsymbol{\mu}}^2 C_{\boldsymbol{\theta}, \boldsymbol{\mu}} \leq \frac{L_{\phi_0}}{2} - \frac{\underline{\eta} \rho_T L_{\boldsymbol{\theta}, \boldsymbol{\mu}}^2}{\gamma_2^2} - (1 + \frac{2}{\underline{\eta} \rho_T}) L_{\boldsymbol{\theta}, \boldsymbol{\mu}}^2 C_{\boldsymbol{\theta}, \boldsymbol{\mu}} =: C_\alpha$ and $\frac{L_{v_z}}{2} + (1 + \frac{2}{\eta \rho_T}) L_{\boldsymbol{\theta}, \boldsymbol{\mu}}^2 C_{\boldsymbol{\theta}, \boldsymbol{\mu}} \leq \frac{L_{v_z}}{2} + (1 + \frac{2}{\underline{\eta} \rho_T}) L_{\boldsymbol{\theta}, \boldsymbol{\mu}}^2 C_{\boldsymbol{\theta}, \boldsymbol{\mu}} =: C_\beta$

Consequently, if $C_\alpha, C_\beta > 0$ satisfies $C_\alpha \leq \min \left\{ \frac{\eta \rho_T}{4}, \frac{1}{4C_\alpha} \right\}$ and $C_\beta \leq \min \left\{ \frac{\eta \rho_T}{4}, \frac{1}{4C_\beta} \right\}$, it holds that $\frac{L_{\phi_t}}{2} + \frac{\beta L_{\boldsymbol{\theta}, \boldsymbol{\mu}}^2}{\gamma_2^2} + (1 + \frac{2}{\eta \rho_T}) L_{\boldsymbol{\theta}, \boldsymbol{\mu}}^2 C_{\boldsymbol{\theta}, \boldsymbol{\mu}} \geq \frac{1}{4\alpha}$ and $\frac{L_{v_z}}{2} + (1 + \frac{2}{\eta \rho_T}) L_{\boldsymbol{\theta}, \boldsymbol{\mu}}^2 C_{\boldsymbol{\theta}, \boldsymbol{\mu}} \geq \frac{1}{4\beta}$

This completes the proof.

