# OpenReview forum: "Constrained Multi-Objective Optimization"
_ICLR.cc/2025/Conference — Submitted to ICLR 2025_

### Official Review · Reviewer_u3Cc · 2024-10-29

**Soundness:** 2
**Presentation:** 2
**Contribution:** 1
**Rating:** 1
**Confidence:** 4

**Summary:**

In this paper, the authors propose a gradient-based algorithm for solving multi-objective optimization problems: MLM-CMOO. Moreover, under the assumption that both the individual objectives as well as the constraints are convex, the authors provide a proof in which they show the convergence rate of their algorithm to reach a Pareto stationary solution. At last, they conduct an experimental study, in which they evaluate their approach in comparison to two competitors, NSGA-II and PSL.

**Strengths:**

- The paper proposes a novel gradient-based algorithm for constraint MOO.
- The authors provide a proof sketch for the algorithm's convergence rate.

**Weaknesses:**

First of all, the paper title is entirely misleading. "Constrained Multi-Objective Optimization" sounds like a book title, or at least a PhD thesis, in which the entire research field is investigated from various perspectives, such as diverse benchmark problems, algorithmic approaches, visualization methods, performance measures, etc. However, this paper is limited to the gradient-based part of that research field and essentially focusses on the introduction of a new algorithm.
Within this work it is stated that the goal of MOO is to find a Pareto stationary solution. However, as the optima of MOO problems usually consist of entire solution sets, the goal of algorithms usually is to find a good approximation of the solution set. Claiming that MOO focusses on finding a single solution would downplay the complexity of this research area.
The proposed approach has been designed for MOO problems that are (strongly) convex. Yet, it is not clear whether this property relates to the search or objective space.
The usefulness of MLM-CMOO remains unclear, as real-world problems likely aren't compositions of purely (strongly) convex single-objective functions. As has been shown in various publications, concatenations of convex problems are a very special case, failing to capture the complexity of most MOO problems. By integrating a single multimodal (single-objective) function, the resulting MOO problem will have several local optima, which serve as traps for gradient-based approaches.
In the benchmark study, the proposed algorithm is compared to NSGA-II and PSL. However, NSGA-II is a population-based approach that evaluates multiple solutions per iteration. In consequence, it will likely take more time to converge. Also, population-based approaches such as NSGA-II are designed to find optima in complex black-box problems. If the problem is convex, gradient-based approaches will usually win such a comparison. Therefore, fair benchmark studies should consider state-of-the-art gradient-based approaches.
The convergence rate is indicated in relation to T, however, T is not specified/defined within the paper.
For citing, in multiple cases the wrong citing command has been used. For instance, it should be "The NSGA-II (Deb et al., 2002)" instead of "The NSGA-II Deb et al. (2002)".
The experimental analysis is kept extremely brief, providing hardly any insights.

**Questions:**

Does the assumption of a convex problem refer to the objective or search space?
According to Assumption 4.1, the individual objectives f_i need to be convex. Isn't this a very extreme assumption, which severely limits the contribution of this work as hardly any MOO problems will consist exclusively of convex components?
Within the experiments, MLM-CMOO has been benchmarked agains NSGA-II and PSL. How does MLM-CMOO perform in comparison to algorithms utilizing similar concepts or other gradient-based approaches? For instance, gradient ascend and gradient sliding methods could provide a fairer comparison.
Is T referring to the number of function evaluations, generations, runtime, ...?
In the literature, you find various works that investigate multimodality in MOO and also define various properties. Is the term "Pareto criticality" identical to a multi-objective "local optimum"? If so, why would you give it a different name?
The work emphasizes its focus on constrained MOO. However, why are constraints necessary in this context? Wouldn't this work look similar, if the problems would be unconstrained?
Table 1 measures the performance until a Pareto stationary point is reached. Again, this looks like a very special case, as one of the main challenges of MOO usually is to find a good approximation of the entire Pareto set (or front, respectively).
How often is each of the three algorithms executed per test problem?

---

### Official Review · Reviewer_v7Z2 · 2024-11-03

**Soundness:** 2
**Presentation:** 2
**Contribution:** 1
**Rating:** 3
**Confidence:** 4

**Summary:**

This paper describes a constrained multi-objective optimisation approach. Instead of using ordinary gradient based algorithms, this paper proposes a constrained gradient based one based on multiple gradient descent algorithms. The new approach seems to have a good convergence to Pareto solutions, and produced good results.

**Strengths:**

MOO is a good topic and approach to processing multiple potentially conflicting objectives. This paper develops a new method toward this direction, which is good.

The paper also provides an convergence analysis, which seems to be theoretically demonstrating the the proposed algorithm can be converged.

**Weaknesses:**

There are a number of weaknesses.

1. Why do you consider gradient based approaches only in this paper to tackle multi-objective optimisation tasks? Under what conditions, gradient based methods are good?

2. The number of objectives in all the datasets should be explicitly identified and described clearly.

3. In your experiments, you compared the proposed method with NSGA-II and PSL. Why didn't you consider SPEA2 and MOEA/D?

4. The results are far from complete --- only Table 1 and Figure 1 are presented. Many interesting results such as the Pareto-fronts for each dataset, the hyper-volume or IGD comparisons are missing. It is not clear to me the comparison between those algorithms are fair. Only providing the loss and time values is not sufficient to understand the effectiveness of the proposed method.

**Questions:**

1. How many independent runs have you carried our for each of the comparison algorithm? All those methods including those gradient based methods and NSGA-II are stochastic ones, different runs with the same parameter setup will produce different results.

2. Have you considered other performance evaluation measures, such as HV and IGD, comparison of Pareto-fronts?

3. Ablation studies: your proposed method has several components. Which one plays more important roles than others? Are all of them useful?

It seems that you submitted this paper in a hurry without completing it.

---

> ### Author Response · Authors · 2024-12-02
>
> Thank you for your comments.
>
> For comment 1, gradient descent (GD) generally outperforms intuitive algorithms in terms of speed of convergence. In addition, GD performs better in large-scale computation and/or distributed computation. Furthermore, although some gradient-free algorithms can get some performance guarantees under some assumptions, most gradient-free algorithms can't provide such guarantees, however, nearly all gradient-based algorithms can provide convergence guarantees.
>
> For comment 2, we restate the number of objectives in all the datasets in the part of the experiment.
>
> For comments 3 & 4, we didn't consider SPEA2 and MOEA/D due to our inadequate reviews and studies, we will add them as the baseline in future work. We will perform more experiments as suggested.
>
> For question 1, we set the same number of independent runs as 5. The result is an average of all runs.
>
> For question 2, we didn't consider other performance evaluation measures, but we worked on them for the next version.
>
> For question 3, the proposed method can be divided into 2 parts, GD to find solution and constraint control. These 2 parts combine to make our algorithm find the CMOO solutions under constraints. Both parts are important for our method.

---

### Official Review · Reviewer_DuGT · 2024-11-03

**Soundness:** 3
**Presentation:** 2
**Contribution:** 2
**Rating:** 3
**Confidence:** 3

**Summary:**

This paper proposes a gradient-based method for constrained multi-objective optimization, named MLM-CMOO. It is proved that the proposed method guarantees a convergence rate of $O(1/\sqrt{T})$. An empirical study demonstrates the effectiveness of the proposed method.

**Strengths:**

1. This paper is well-written and easy to follow.
2. The technical and mathematical details are clearly presented. The proofs are well-organized and seem correct.
3. This paper is novel and original, since constrained gradient-based MOO has not been well-studied.

**Weaknesses:**

1. In some parts of the text, "multi-objective optimization" is used, while in others, "multiple-objective optimization" is used. It is recommended that the author standardize the terminology.
2. The authors claim that the proposed method outperforms the state-of-the-art. It might be somewhat controversial to regard NSGA-II as state-of-the-art as it was proposed 20 years ago.
3. L38. The authors believe that gradient-free methods can not provide a convergence guarantee. I think gradient-free methods could also have a convergence guarantee under certain assumptions [1,2].
4. L51, "However, this transformation can not give a stable guarantee for the convergence rate as it may give the farthest Pareto front for the given coefficient." What does "farthest Pareto front" mean? Could you please provide some clarification for this claim or some references?
5. I think the title does not accurately reflect the contributions of this paper. The author could consider using "Gradient-based Constrained Multi-Objective Optimization" or "Constrained Multiple-Gradient Descent".
6. L397. The authors believe that NSGA-II cannot handle constraints. Actually, NSGA-II is originally designed to solve constrained problems [3].
7. The empirical study is not well-designed. The proposed method is compared with two black-box methods on deep learning tasks. I believe it is more reasonable to use gradient-based methods as baselines, for instance, [4, 5]. Some experimental details are missing, e.g., the population size of NSGA-II.
8. From Fig. 1, I did not find that CMOO significantly outperforms the baselines. It is recommended to use some performance indicators to quantitatively measure the difference.
9. L406. Typo: $1e^{-5}$ --> $10^{-5}$.

References

[1] Zheng, Weijie, Yufei Liu, and Benjamin Doerr. "A first mathematical runtime analysis of the Non-Dominated Sorting Genetic Algorithm II (NSGA-II)." AAAI 2022.

[2] Do, Anh Viet, et al. "Rigorous runtime analysis of MOEA/D for solving multi-objective minimum weight base problems." NeurIPS 2023.

[3] K. Deb, A. Pratap, S. Agarwal, and T. Meyarivan. A fast and elitist multiobjective genetic algorithm: NSGA-II. IEEE Trans. Evol. Comput., 2002.

[4] Liu et al., Profiling Pareto front with multi-objective stein variational gradient descent, NeurIPS 2021.

[5] Chen et al., Multi-Objective Deep Learning with Adaptive Reference Vectors, NeurIPS 2022.

**Questions:**

Please see "Weaknesses".

---

### Official Review · Reviewer_jZrr · 2024-11-04

**Soundness:** 1
**Presentation:** 1
**Contribution:** 2
**Rating:** 3
**Confidence:** 3

**Summary:**

This work proposes a gradient-based optimization algorithm, MLM-CMOO, to solve constrained multi-objective optimization (CMOO) problems. The authors conduct a convergence analysis and several experiments to demonstrate the effectiveness of the proposed MLM-CMOO algorithm.

**Strengths:**

The authors conduct a convergence analysis for MLM-CMOO and provide several theoretical proofs in the supplementary materials.

**Weaknesses:**

1. The submission focuses on constrained multi-objective optimization (CMOO), however, no CMOO algorithms are compared in the experiments. The authors should compare several CMOO algorithms in their experiments. For example, the authors reviewed some gradient-free CMOO algorithms in the related work section, which could be included in the comparisons.
2. In Section 2, only gradient-based MOO and gradient-free CMOO are discussed in the related work. The authors should also review some constraint handling techniques (CHTs).
3. The experiments in Section 5 are all multi-task learning (MTL) problems. The authors should add explanations to clarify how these MTL problems are used to evaluate the performance of CMOO algorithms, including what the constraints are in these MTL problems.
4. The presentation of experimental results lacks detail. The authors should provide detailed explanations of Table 1 and Figure 1 to help readers understand experimental results. For example, the caption for Figure 1 is too brief, the authors should clarify the meaning of each subfigure in Figure 1.
5. The symbols in the submission are inconsistent. For instance, the $i^{th}$ objective function is denoted as $f_i(x)$ or $f^i(x)$.

**Questions:**

In line 132, why do the authors state that the goal of MOO is to find a Pareto optimal solution? In fact, for many real-world MOO applications, the goal is to find a set of well-distributed Pareto optimal solutions.

---

> ### Author Response · Authors · 2024-12-02
>
> Thank you for the time to review our paper.
>
> For comments 1 & 2, We added a new paragraph to compare gradient-free CMOO and provide a review of CHTs. Details are a simple taxonomy of the constraint handling methods in nature-inspired optimization algorithms: Penalty functions, decoders, Special operators, and Separation techniques. Several types of penalty functions are used with evolutionary algorithms (EAs), the most important ones include [12] Death penalty, Dynamic penalty, Static penalty, Adaptive penalty, and Stochastic ranking. As an example of decoders, [5] proposed a homomorphous mapping (HM) method between an n-dimensional cube and feasible space. The feasible region can be mapped onto a sample space where a population-based algorithm could run a comparative performance [5, 9-11]. However, this method requires high computational costs. A special operator is used to preserve the feasibility of a solution or move within a special region [6–8]. Nevertheless, this method is hindered by the initialization of feasible solutions in the initial population, which is challenging with highly constrained optimization problems. Unlike the penalty function technique, another approach exists that separates the values of objective functions and constraints in the nature-inspired algorithms (NIAs) [13], which is known as the separation of objective function and constraints. The authors of [14] initially proposed the idea of dividing the search space into two phases. In the first phase, feasible solutions are found, and optimizing the objective function is considered in the second phase. Representative methods of this type of CHT are the Constraint dominance principle (CDP), Epsilon CHT, and Feasibility rules.
>
> For comment 3, a typical MTL system is given a collection of input points and sets of targets for various tasks per point. A common way to set up the inductive bias across tasks is to design a parametrized hypothesis class that shares some parameters across tasks. One effective solution for MTL is finding solutions not dominated by others, which is the same objective of MOO problems. Such solutions are said to be Pareto optimal.
>
> For comments 4 & 5, we worked on more experiments and revised inconsistent errors. We polished our paper and avoided such errors.
>
> For question 1, some previous works work on getting a set of Pareto optimal solutions, however, most gradient-based MOO works focus on finding a Pareto optimal or stationary point at a fast speed, such as [1-4]. As the proposed method is also a gradient-based MOO work, so we set the same goal as previous works. In addition, if we want to get a set of Pareto optimal solutions, we can use some different initial points of the proposed algorithm to get the different Pareto optimal solutions.
>
> 1.S. Zhou, et al, On the Convergence of Stochastic Multi-Objective Gradient Manipulation and Beyond
>
> 2.H. Yang, et al, Federated Multi-Objective Learning
>
> 3.S. Liu, et al,The stochastic multi-gradient algorithm for multi-objective optimization and its application to supervised machine learning, arxiv 2021.
>
> 4.O. Sener , et al, Multi-Task Learning as Multi-Objective Optimization
>
> 5.S. Koziel, et al, A decoder-based evolutionary algorithm for constrained parameter optimization problems
>
> 6.Z. Michalewicz, Genetic algorithms+ data structures=evolution programs
>
> 7.M. Schoenauer, et al, Evolutionary computation at the edge of feasibility.
>
> 8.M. Schoenauer, et al, Boundary Operators for Constrained Parameter Optimization Problems
>
> 9.D.G. Kim, et al, Riemann mapping constraint handling method for genetic algorithms.
>
> 10.D.G. Kim, et al, Landscape changes and the performance of Mapping Based Constraint handling methods.
>
> 11.S. Koziel, et al, Evolutionary algorithms, homomorphous mappings, and constrained parameter optimization.
>
> 12.O. Kramer, A review of constraint-handling techniques for evolution strategies
>
> 13.D. Powell, et al, Using genetic algorithms in engineering design optimization with non-linear constraints
>
> 14.R. Hinterding, et al, Your brains and my beauty: parent matching for constrained optimization

---

> > ### Comment · Reviewer_jZrr · 2024-12-03
> >
> > Thanks for the authors' rebuttals.
> >
> > I noticed that the authors have reviewed some related works on CHTs. This addresses my second concern to some extent.
> >
> > However, my concerns 1, 3, and 4 remain unaddressed: The authors did not include any new CMOO algorithms as comparisons in their experiments, and it is still unclear what the constraints are in MTL problems. Additionally, experimental details are still missing. Therefore, I will keep my score.

---

### Meta-Review · Area_Chair_kyVU · 2024-12-18

**Metareview:**

The paper proposes a new gradient-based method, MLM-CMOO, for constrained multi-objective optimization (CMOO). It offers a convergence guarantee to Pareto stationary solutions with a rate of O(1/sqrt(T)). However, the paper's title is misleading as it focuses solely on a single algorithm rather than providing a broad overview of CMOO. The assumptions of convexity for objectives and constraints are overly restrictive, limiting the algorithm's applicability to real-world problems. The experimental setup compares MLM-CMOO with NSGA-II but lacks sufficient details and rigor, making it difficult to assess the algorithm's performance. The paper's contribution is considered limited due to these issues, and it is hence recommended for rejection.

**Additional Comments On Reviewer Discussion:**

The authors provided limited responses during the discussion period and most of the concerns raised by the reviewers remain.

---

### Decision · Program_Chairs · 2025-01-22

Reject